# Tracking HIV-1 recombination to resolve its contribution to HIV-1 evolution in natural infection

Hongshuo Song[1,14], Elena E. Giorgi[2], Vitaly V. Ganusov[3], Fangping Cai[1], Gayathri Athreya[4], Hyejin Yoon[2], Oana Carja[5], Bhavna Hora[1], Peter Hraber [2], Ethan Romero-Severson[2], Chunlai Jiang[1,6], Xiaojun Li[1], Shuyi Wang[7], Hui Li[7], Jesus F. Salazar-Gonzalez[8,15], Maria G. Salazar[8], Nilu Goonetilleke[9], Brandon F. Keele[10], David C. Montefiori[1], Myron S. Cohen[9], George M. Shaw[7,11], Beatrice H. Hahn[7,11], Andrew J. McMichael[12], Barton F. Haynes[1], Bette Korber[2], Tanmoy Bhattacharya[2,13] & Feng Gao [1,6]

Recombination in HIV-1 is well documented, but its importance in the low-diversity setting of within-host diversification is less understood. Here we develop a novel computational tool (RAPR (Recombination Analysis PRogram)) to enable a detailed view of in vivo viral recombination during early infection, and we apply it to near-full-length HIV-1 genome sequences from longitudinal samples. Recombinant genomes rapidly replace transmitted/ founder (T/F) lineages, with a median half-time of 27 days, increasing the genetic complexity of the viral population. We identify recombination hot and cold spots that differ from those observed in inter-subtype recombinants. Furthermore, RAPR analysis of longitudinal samples from an individual with well-characterized neutralizing antibody responses shows that recombination helps carry forward resistance-conferring mutations in the diversifying quasispecies. These findings provide insight into molecular mechanisms by which viral recombination contributes to HIV-1 persistence and immunopathogenesis and have implications for studies of HIV transmission and evolution in vivo.

[1] Duke Human Vaccine Institute and Department of Medicine, Duke University Medical Center, Durham, NC 27710, USA. [2] Theoretical Division, Los Alamos National Laboratory, Los Alamos, NM 87544, USA. [3] Department of Microbiology, University of Tennessee, Knoxville, TN 37996, USA. [4] Office for Research & Discovery, University of Arizona, Tucson, AZ 85721, USA. [5] Department of Biology, University of Pennsylvania, Philadelphia, PA 19104, USA. [6] National Engineering Laboratory For AIDS Vaccine, College of Life Science, Jilin University, Changchun, Jilin 130012, China. [7] Department of Medicine, University of Pennsylvania, Philadelphia, PA 19104, USA. [8] Department of Medicine, University of Alabama at Birmingham, Birmingham, AL 35294, USA. [9] Departments of Microbiology and Immunology & Medicine, University of North Carolina at Chapel Hill, Chapel Hill, NC 27599, USA. [10] AIDS and Cancer Virus Program, Frederick National Laboratory for Cancer Research, Frederick, MD 21702, USA. [11] Department of Microbiology, University of Pennsylvania, Philadelphia, PA 19104, USA. [12] Weatherall Institute of Molecular Medicine, University of Oxford, Oxford OX3 9DS, UK. [13] Santa Fe Institute, Santa Fe, NM 87501, USA. [14]Present address: United States Military HIV Research Program, Walter Reed Army Institute of Research, Silver Spring, MD 20910, USA. [15]Present address: MRC/UVRI and LSHTM Uganda Research Unit, Plot 51-57, Nakiwogo Road, Entebbe, Uganda. These authors contributed equally: Hongshuo Song, Elena E. Giorgi. These authors jointly supervised this work: Bette Korber, Tanmoy Bhattacharya, Feng Gao. Correspondence and requests for materials should be addressed to F.G. (email: fgao@duke.edu)

A recombinant is a genetic sequence that carries regions from two genetically distinct parental strains. That recombination contributed to HIV-1 quasispecies evolution in vivo was observed early after the discovery of HIV[1], and, soon after, recombination's important role in global HIV-1 diversification was established[2,3]. Ninety inter-subtype recombinants have been shown to be recurrent among circulating HIV-1 viruses (for a current listing of these circulating recombinant forms (CRFs), see https://www.hiv.lanl.gov/content/sequence/HIV/CRFs/CRFs.html). Some are very common epidemic lineages like CRF01_AE, which is predominant in Thailand and China, and CRF02_AG, which is common in West Africa[3]. Recombination increases overall genetic complexity of viral populations more than just the accumulation of site mutations alone, thereby raising the likelihood for recombinants to find favorable genetic configurations and facilitating faster adaptation[4]. Recombination is believed to contribute to viral diversity and fitness[5–7], drug resistance[8,9], immunological escape[10,11], and disease progression[7,12].

Estimates of recombination rates in vivo vary among different studies—$1.4 \times 10^{-5}$ to $2 \times 10^{-4}$ breakpoints per site per generation[13–15]—comparable to the point mutation rate of $2.2–5.4 \times 10^{-5}$ per base per generation[16,17]. In vitro studies have found HIV-1 to be a highly recombinogenic virus[18,19]. However, in order to detect recombinants, these studies utilize "foreign" gene inserts or sequences from divergent subtypes, unlike in vivo infections. Schlub et al.[20] describe an in vitro system that better mimics the typical in vivo scenario, and observe that recombination occurs non-randomly, with a frequency of the order of $10^{-3}$ breakpoints per nucleotide per round of infection, with multiple template switches between parental strains.

A number of factors have limited a better understanding of the role of recombination in studies that looked at in vivo HIV-1 infection. PCR-derived recombination artifacts are common among sequences generated by bulk PCR products and, when not excluded, may have affected older recombination studies that did not use single-genome amplification (SGA)[21]. Similarly, studies that utilize sequences sampled only in chronic infection may not be able to resolve ambiguities between putative parents and products of recombination[14,15]. Other studies analyzed only a small portion of the viral genome. Such shortcomings can bias estimates of the frequencies and evolutionary dynamics of recombinants in vivo.

Many recombination detection tools currently available focus on detecting recombinants between genetic subtypes and are best applied to highly diverse sequences. Based on a sliding window approach, RIP[22] was the first bioinformatics tool to automatically screen for inter-subtype recombinants. More recent and advanced approaches are available today for detection of recombinants and their breakpoints within a single alignment. RECCO[23] finds nominally significant recombinants by comparing the cost of obtaining each sequence either by mutation or by recombination. It does not, however, list parental strains or enumerate distinct recombination events. Among the programs that provide a list of detected recombinants, including their potential breakpoints and parental strains, is the suite RDP4[24], which, in addition to implementing the recombination detection program (RDP) approach, includes several other existing tools, providing corroboration of findings across the different methods. While these tools range in methods and strategies, the RDP test itself is based on phylogenetic methods, and, as Posada and Crandall[25] showed, in low-divergence scenarios like the recent infections presented in this study, these methods have less power to detect recombinants.

Defining the frequency of recombinants and identifying breakpoints is particularly important in phylogenetic analyses because recombination significantly biases estimates of mutation rates and confounds relationships in standard phylogenetic tree reconstructions[26]. Currently, the bioinformatics tool GARD[27] addresses this issue by finding likely breakpoints of recombination within the alignment and reconstructing a series of phylogenetic trees of genetic segments within the breakpoints. However, this program does not provide a list of recombinants or sequence relationships found within the alignment, and therefore is not helpful when analyzing low-diversity and/or shorter regions, where eliminating a few sequences yields more statistical power than breaking up the alignment.

Our bioinformatics tool, called Recombination Analysis PRogram (RAPR), is based on the Wald–Wolfowitz Runs Test[28] and allows for computationally efficient detection of recombinants. While variations on the Runs Test have been employed in the past to detect recombination in the context of gene conversion[29], this method has not been implemented on more recent and faster computational platforms.

Here, we show that RAPR is sensitive to identification of recombinant stretches in low-diversity settings, where only a small number of distinguishing bases are involved, and has the capability to discern parental strains from recombinants, in part by accounting for longitudinal sampling times. Moreover, it can address the lineage structure of the alignment by differentiating between the initiating recombinant event and its mutated descendants, when these are known for analysis. As a result, RAPR is able to reconstruct a hierarchy of serial recombination events as recombinants of recombinants emerge over time. We first applied RAPR to a dataset of 3260 5′ and 3′ half genome sequences derived by SGA from 9 participants infected with multiple transmitted/founder (T/F) viruses and sampled longitudinally over time. Because all infections were initiated by multiple, genetically distinct T/Fs, recombinants between the infecting strains could be readily identified and tracked. This offered a unique opportunity to investigate how recombination impacts in vivo viral evolution and immune escape. We next applied RAPR to sequences from an individual (CH0505) whose infection was established by a single T/F virus and whose immune response has been extensively studied[30]. We used this example to illustrate how to apply RAPR in a single T/F setting and illustrate the role of recombination in carrying forward resistance mutations in a complex quasispecies. While heterogeneous infections have been analyzed before to estimate recombination rates[31], and longitudinal samples have been used to estimate recombination breakpoints[32], this is the first study to estimate how both recombination rates and the accumulation of breakpoints contribute to the evolution of the HIV-1 quasispecies over time during natural infection.

## Results

**Study participants.** Previous studies have shown that approximately 80% of heterosexually transmitted HIV-1 infections are initiated by a single T/F virus and only 20% are due to multiple, genetically distinct T/F viruses[33]. In the latter case, due to the genetically distinct quasispecies coevolving in the host, it becomes easier to follow the history of recombination from the beginning of the infection. To this purpose, we distinguished participants productively infected with more than one T/F virus (heterogeneous infection) from those infected with a single T/F virus (homogeneous infection) by characterizing patterns of sequence diversity at the earliest time point (Fig. 1, Table 1, and Supplementary Figs. 1 and 2) using statistical modeling, phylogenetic trees, and highlighter plots, as previously described[33]. It is important to note that the number of infecting strains, the incidence of superinfection, and the estimated number of days since infection play no role in the detection of recombination, except

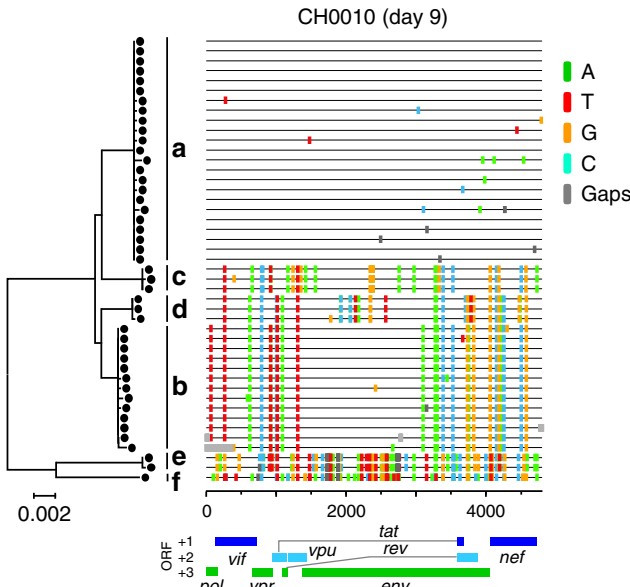

**Fig. 1** Highlighter plot and phylogenetic tree of 3′ half genome sequences at screening time for CH0010. Six distinct T/F viruses (labeled a–f) were identified. Each line represents a 3′ half genome sequence, and mutations from the major T/F (first sequence at the top) are color coded according to nucleotide

**Table 1 Fiebig stage, transmission route, days since infection, and number of T/F viruses of heterogeneously infected individuals**

| Subject | Fiebig stage | Transmission route | Days from infection (95% CI) | No. of T/F viruses | |
|---|---|---|---|---|---|
| CH0010 | I/II | Heterosexual | 9 (7, 11) | 3 | 6 |
| CH0078 | I/II | Heterosexual | 11 (7, 14) | 2 | 2 |
| CH0200 | I/II | Heterosexual | 15 (13, 17) | 3 | 5 |
| CH0047 | III | Heterosexual | 25 (21, 29) | 2 | 2 |
| CH0228 | III | Heterosexual | 24 (21, 28) | 2 | 3 |
| CH0275 | I/II | Heterosexual | 12 (8, 16) | 1 | 2 |
| CH0654 | I/II | MSM | 15 (13, 17) | 9 | 8 |
| CH1244 | IV | Heterosexual | 12 (11, 13) | NA | 2 |
| CH1754 | III | Heterosexual | 14 (12, 15) | 6 | 7 |

Days from infection were calculated using previously published methods[33]. These methods generally strongly correlate with Fiebig stage, though occasional deviations still happen, especially when under strong selection. In our present dataset, CH1244 is one such exception, with a Fiebig stage IV yet diversity that would be expected—under a model of random accumulation of mutation—to be roughly 2 weeks into the infection
MSM men who have sex with men

that we do not count the putative founder strains as recombinants since we are focusing on recombination in the recipient rather than in the donor. However, calculating the time since infection allows for aligning dynamics of recombination and viral loads to facilitate comparisons. Participant selection and characteristics, T/F identification, longitudinal sampling, and timing of infection are described in the Methods.

**The RAPR program**. To characterize recombination events and determine the frequency of recombinants over time, we developed the RAPR tool, available as a web-based interface in the LANL database (http://www.hiv.lanl.gov/content/sequence/RAP2017/rap.html). The program relies on the fact that recombination events can be identified through the inheritance of blocks of DNA that carry distinguishing bases. Therefore, it finds recombinants by comparing every set of three sequences in the alignment and applying the Wald–Wolfowitz Runs Test[28] to check for randomness in the distribution of variable positions, identifying regions in a sequence that are significantly more similar to one candidate parent than another. It then classifies the viral population into clusters of T/F viruses and their descendants—which present mutations explainable by randomly distributed base changes—or identified recombinants and their descendants. In the latter case, RAPR also determines the likeliest parents, based both on the clustered distribution of sequence differences and on their time of sampling, and delineates regions in which the breakpoints are likely to lie.

As an example of RAPR performance and output, we illustrate detailed results for participant CH0010, infected with 6 sampled T/F viruses, labeled a–f (Figs. 1 and 2a). Out of 103 unique sequences from 4 time points (days 9, 26, 72, and 188 post infection), RAPR detected 52 distinct de novo recombinants, 4 of which gave rise to 11 descendants (Fig. 2). By day 72 onward, all sequences were either recombinants or descendants of recombinants (Fig. 2). Estimated breakpoint intervals (illustrated as open boxes in Fig. 2, shaded in gray when the breakpoint is statistically significant, see Methods) tend to be larger and sparser earlier,

when sequences are less diversified. Recombinant sequences accumulated more breakpoints over time as a result of continuous serial recombination between earlier recombinants (Fig. 2b). Similar results were obtained on both 5′ and 3′ half genome sequences from all 9 participants (Supplementary Figs. 3–11). The frequency of recombinants and their descendants increased in all individuals, eventually replacing all T/F lineages. Complete replacement with recombinants of all distinctive T/F lineages that were evident in the earliest available time point was observed as early as 35 days, and typically was evident within 100 days (Fig. 3). The exception was participant CH0275, who was sparsely sampled, with no time points available between 12 and 187 days; all CH0275 sequences sampled at day 187 were, however, recombinant forms (Fig. 3a).

**Estimations of number of breakpoints**. Template switches occurring in nearly identical regions of the genome are likely to go undetected. Cromer et al.[31] modeled this phenomenon and estimated between 5 and 14 template switches per recombinant genome. In this study, we observed a high frequency of recombination breakpoints in our data when both significant and non-significant but potential breakpoints were considered, consistent with what was reported by Cromer et al.[31]. However, because of regions of near identity between the parents, the significant breakpoint counts of RAPR are inevitably lower than the true number of template switches, as illustrated in Supplementary Fig. 12. In order to quantify how many undetected breakpoints resulted from template switches in regions of high homology between parental strains, we used a recombination simulation strategy (see Methods for details). As expected, RAPR consistently underestimated the true number of breakpoints (Supplementary Figs. 13 and 14), and yielded a distribution of significant breakpoints lower than the actual number of breakpoints introduced in the simulation. This is not surprising given that all sets of sequences used in the simulation had very low within-alignment diversity (the within T/F mean Hamming distance range per nucleotide per sequence was 0.76–2.93% across all parental pairs), and many of the random recombination breakpoints in our simulation occur in sequence stretches where both parents were identical or nearly identical. As a result, this study alone cannot bound the recombination rate from above, and Cromer et al.[31] values are consistent with the lower bounds we obtain.

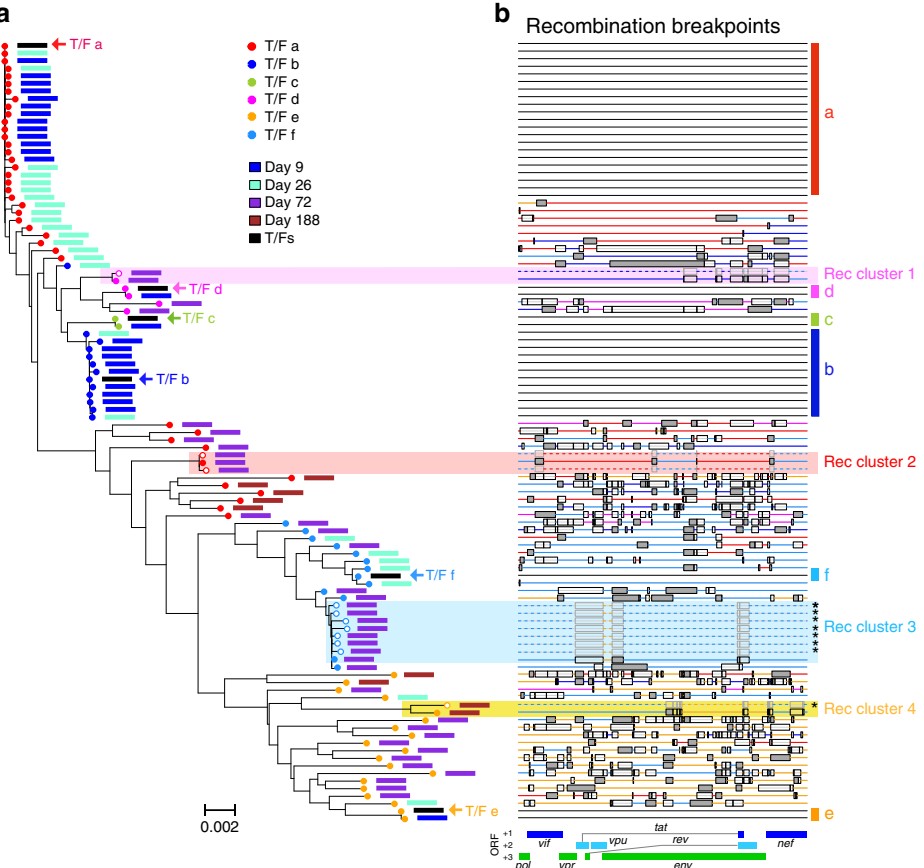

**Fig. 2** Identification of recombinants by RAPR. **a** Phylogenetic tree of 3′ half genome sequences from longitudinal samples from CH0010. Shaded regions show clonal expansion events and are color coded to indicate their closest T/F lineage. In each cluster, solid circle represents the originating (de novo) recombinant for that cluster and open circles represent its recombinant descendants. **b** Recombination breakpoints. Each line represents a sequence, and colored lines represent recombinants. Colored intervals in each recombinant sequence indicate the parental T/F lineage. Black boxes indicate the interval where the breakpoints are most likely to have occurred, and when the breakpoints are statistically significant, the box is shaded in gray. Shaded regions show clonal expansion events and are color coded to indicate their closest T/F lineage. Dashed lines indicate recombination descendants derived from a particular de novo recombinant as shown as a solid line in the same shaded group. The black stars on the right indicate the sequence lineage that RAPR identified as descendants of a single recombination event, while the program RECCO failed to recognize it as such

To obtain a posterior distribution on the number of true breakpoints, we also implemented a simulation based on an approximate Bayesian computation method[34] and applied it to the same early time samples mentioned above (see Methods for details). We conclude that for all recombinants for which RAPR detected 2 significant breakpoints, we are 95% confident that the true breakpoints were no less than 2. A similar statement holds when RAPR detected 4 breakpoints, with one exception (Supplementary Table 1). As the number of breakpoints increases, they are more likely to go undetected: see for example two of the recombinants from CH0228 3′ half sequences, for which RAPR detects 5 and 6 significant breakpoints, but the 95% confidence limit (CL) is 6 and 9 respectively. Supplementary Fig. 15 illustrates how, as the number of template switches increases, more breakpoints fall within regions of parental homology, making it harder to accurately detect the true number of breakpoints. In conclusion, our tests indicate that RAPR provides a reasonable, though negatively biased, estimate for the true number of breakpoints; and the bias is small when the true number of breakpoints is small.

**Comparison with existing recombination detection tools.** Despite the bias described above, our program proves useful in low-diversity scenarios. A good way to test this is using

biologically sound simulated datasets where the exact recombinants and their breakpoints are known. To realize this, we randomly generated sets of 100 artificial recombinants with known crossover points, each from three different pairs of natural strains that carried 0.6%, 1%, and 1.2% relative diversity, respectively (see Methods). In the lowest (0.6%) diversity setting, RAPR identified 77 out of 100 unique recombination events and 16 recombination descendants, missing 6 out of 100 known events (Supplementary Table 2). The other computational tools we used to test on the same simulated datasets detected overall less recombination events than RAPR (Supplementary Table 2). However, we should note that selection could impact our results when a few nearby mutations localized proximally are co-selected. Therefore, it is possible that generating recombinants in other ways, e.g., by including some type of selection when sampling sequences, could reduce the efficiency of RAPR at detecting recombinants.

For further comparisons, we tested different recombination detection tools on CH0010, the heterogeneous participant we used above to illustrate the RAPR output. RAPR identified 52 recombination events in this dataset, while the other tested tools detected between 2 and 18 (Fig. 2b) when accounting for sampling times. One feature of RAPR is that it can identify clusters of sequences that likely descended from a single recombination event. When a single recombination event leads to a lineage, i.e., a cluster of sequences that differ from one

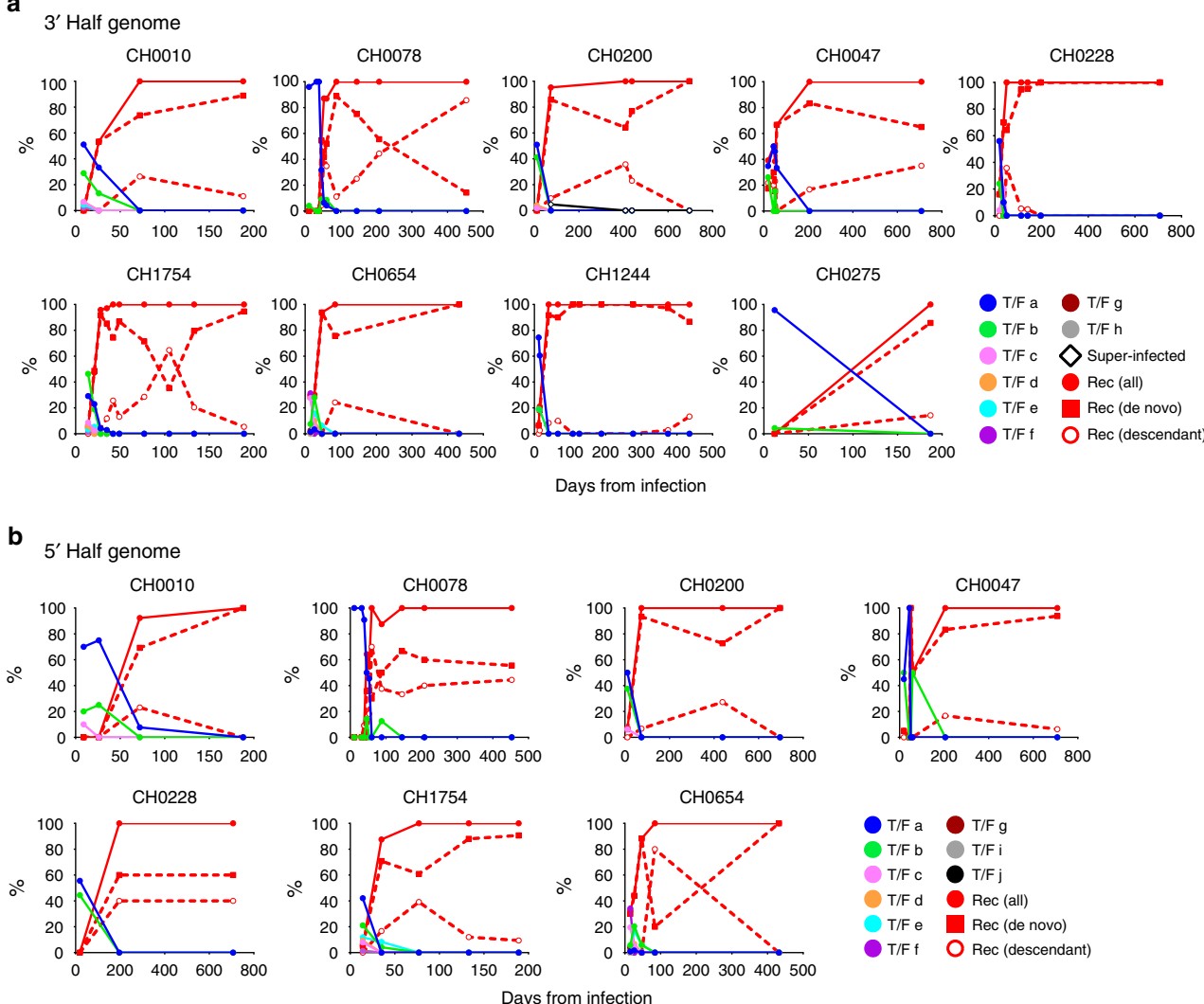

**Fig. 3** Proportion of recombinants and their descendants in the viral population over time. The 3′ half genome (**a**) and 5′ half genome (**b**) sequences were analyzed separately. At each time point, colored dots indicate the percentages of T/F sequences and their lineages, red dots indicate all recombinants, red squares de novo recombinants, red circles descendants of recombinants, and open diamonds superinfection variants

another by a small number of random mutations, many recombination detection tools classify each resulting sequence as a recombinant rather than identifying the entire cluster as a lineage from a single recombinant founder strain. As a result, recombination rates can be overestimated. RAPR avoids over-counting descendants as independent recombination events by breaking the alignment into clusters that are likely to have originated either via mutation from one of the original T/Fs or from a single recombination event and assigning a single "founder" (either a T/F or the original recombination event) for each cluster (Fig. 4).

RAPR assumes that parental sequences are not sampled later than the recombinant, and therefore when resolving parental strains versus recombinants, the program does not allow later time point sequences to be parents of earlier recombinants. However, one needs to be careful when interpreting results in that some earlier lineages may in fact be latent or escape sampling and then reappear at later time points. When in doubt, the user should consider multiple runs, with and without specifying sequence time points. In our study, we make the simplifying assumption that the sampling at each time point is adequate, so that sequences close to the parents are expected to be already

sampled at this or earlier time points. Furthermore, excluding later time points as parentals vastly reduces the multiple-testing corrections needed.

**In vivo recombination rate**. To investigate potential mechanisms for the rapid loss of T/F viruses and their lineages during early infection, we developed a mathematical model of HIV-1 evolution that includes recombination. The model considers cells that can be uninfected, infected with one viral variant, or coinfected with multiple variants (Supplementary Fig. 16), and explores the impact of coinfection and selection on the prevalence of recombinants over time. In the absence of selective advantage of recombinants, a moderate frequency of cell coinfection by two different viral variants (on average 3.3%, range 0.7–8.0% among the 9 participants) was sufficient to explain the rapid accumulation of recombinants and the loss of T/F viruses when 3′ half genome sequences were analyzed (Table 2 and Supplementary Fig. 17a). Modeling 5′ half genome sequences showed similar results (Supplementary Table 3 and Supplementary Fig. 17b). Alternatively, accumulation of recombinants could also be explained by their moderate selective advantage over T/F viruses,

ranging from 0.034 per day to 0.5 per day. Either mechanism could predict the time at which 50% of the viral population had been replaced by recombinants; for example, the "coinfection" model estimated that the median half-time ($T_{1/2}$) across 9 participants was 27 days for 3′ half genome sequences (Table 2). Similar results were obtained with 5′ half genome sequences (Supplementary Table 3).

Deeper understanding of the relative contribution of these two mechanisms in accumulation of recombinants can be gleaned from a detailed analysis of recombination: if the replacements were driven by a selective advantage of a few particular

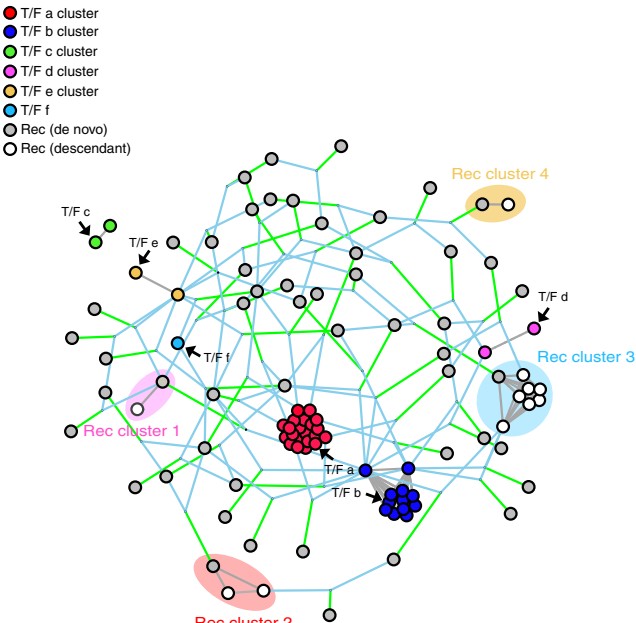

**Fig. 4** Clustering graph of the 3′ half genome sequences from CH0010. Each circle represents a sequence, and circles of the same color represent a T/F lineage. Gray circles represent de novo recombinant sequences and open circles represent descendants of recombinant sequences. Clusters are calculated using a greedy algorithm and they originate either from a T/F lineage or through recombination across lineages. Recombination clusters are shaded and color coded to indicate their closest T/F lineage. At most one sequence per recombination cluster is called a de novo recombinant: the rest are assumed to be descendants of this recombinant sequences. Light blue lines point to recombinant parents and green lines to the recombinant child

recombinants, one or few mutational clusters arising from recombinants would be highly represented. Given that selective forces due to immune pressure are subject to change over time, and that new resistant forms of the virus can arise over time, such dominance may be transient. While the replacement of the T/F lineages was most frequently due to accumulation of novel (de novo) recombinants (Fig. 3), there were some exceptional time points in some participants where a newly arisen recombinant lineage dominated the sample, indicative of positive selection acting on a recombinant (Fig. 3). The most striking examples of this were in the 3′ half from CH0078 at day 453, with one recombinant form having 26 descendants dominating the sample (Fig. 3a, Supplementary Fig. 4a), and in CH0654 at day 84, with a recombinant form with 16 descendants dominating the sample (Fig. 3b, Supplementary Fig. 9b). Furthermore, all participants had recombinant forms that were found to be repeated multiple times, with descendent lineages sometimes persisting for weeks (Fig. 3, Supplementary Figs. 3–11). These observations suggest selection may have played a role in transient emergence of some recombinant lineages.

The number of de novo breakpoints observed correlated better with the number of novel mutations than with viral load (VL), consistent with the hypothesis that periods of more intense positive selection may influence both patterns of emergent recombinants and mutation. The coincidence in the timing of periods of peak accumulation of new base substitutions and recombination breakpoints (Fig. 5) was statistically significant for the 3′ half genomes ($p < 0.015$ by one-sided sign test) but not for the 5′ halves. While recombination requires coinfection, novel mutations are independent of it. This suggests that neither of the two simple models described above captured the full in vivo dynamics observed in our data, and that there may have been periods of increased selective pressure driving changes in the quasispecies, evident as coincident peaks in de novo recombination and base mutation events in the longitudinal sample. Thus, while occasionally selection can favor the dominance of one or few recombinant forms, diversifying selection can also periodically act to increase the prevalence of both new mutations and recombinants. Therefore, both factors—high VL that enhances coinfection of cells and selection—may have contributed to the rapid appearance of recombinants.

**Comparison with mutation rates.** When combining the RAPR output across all 9 participants, we observed that while the rates of accumulation of new mutations and new observed recombination events were of the same order (~$10^{-5}$ per sequence per nucleotide per day; Fig. 5), the rate of new base mutations at any

**Table 2 Estimated rates of coinfection, the percentage of coinfected cells, and the half replacement time by recombinants during the infection (3′ half genome)**

| Subject | Coinfection rate (/day) | % Coinfected cells | $T_{1/2}$ (days) |
| --- | --- | --- | --- |
| CH0010 | 0.067 (0.051, 0.093) | 3.3 | 23.3 (19.3, 27.6) |
| CH0078 | 0.088 (0.081, 0.095) | 4.2 | 47.3 (44.6, 50.5) |
| CH0200 | 0.056 (0.033, 0.094) | 2.7 | 29.5 (21.9, 41.8) |
| CH0047 | 0.014 (0.008, 0.022) | 0.7 | 39.2 (31.2, 55.1) |
| CH0228 | 0.103 (0.068, 0.153) | 4.9 | 27.0 (24.3, 31.0) |
| CH1754 | 0.149 (0.123, 0.177) | 6.9 | 19.8 (18.9, 21.0) |
| CH0654 | 0.035 (0.027, 0.044) | 1.7 | 27.4 (24.8, 30.6) |
| CH1244 | 0.175 (0.121, 0.27) | 8.0 | 19.5 (16.8, 22.8) |
| CH0275 | 0.024 (0.014, 0.039) | 1.2 | 114.0 (73.9, 172.0) |
| Median | 0.067 | 3.3 | 27.0 |

The rate of coinfection is given as $\gamma\beta I$, and the percentage of coinfected cells during the infection is $F_c = \gamma\beta I/(\gamma\beta I + \gamma\beta T)$ with $\gamma\beta T = 2$/day. $T_{1/2}$ indicates the predicted time when the frequency of recombinants reaches 50% of the viral population. Coinfection rate was estimated by fitting 3′ and 5′ data simultaneously using maximum likelihood (see Methods for more detail). In CH0654, all sequences were recombinants at day 84 before the initiation of ART at day 112. Therefore, ART in CH0654 did not affect the analysis

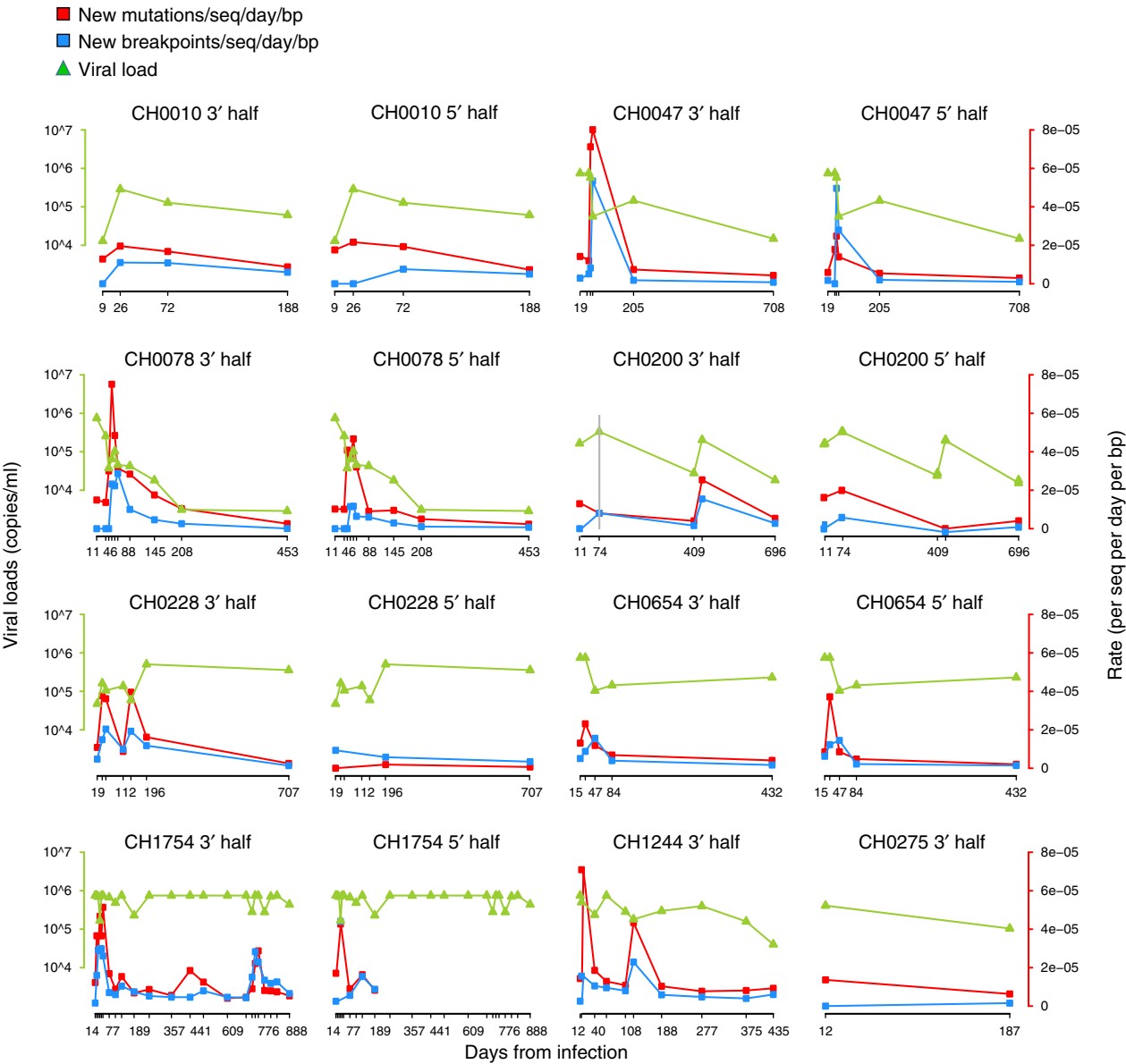

**Fig. 5** Comparison of new breakpoint and mutation rates. Each graph shows the accumulation rate of new breakpoints per sequence per nucleotide per day (blue lines), new mutations per sequence per nucleotide per day (red lines), and log-scale viral loads (green lines) in both half genomes over time for all nine participants. Upper VL detection limit is 750,000 copies/ml. The gray vertical line in CH0200 3′ half highlights the time point at which a superinfecting T/F was observed. The sequences from the last time point (day 432) in CH0654 were excluded from the analysis due to the initiation of ART at day 112

time point (per sequence, per nucleotide, per day) was always higher than the rate of new breakpoints (per sequence, per nucleotide, per day) in both genome halves across all participants, with the only exception of the 5′ half genome in CH0228 (Fig. 5). Both recombination and mutation rates tended to decrease with time. The difference in rates between recombination and mutation was significant ($p = 1.9 \times 10^{-8}$ and $p = 1.4 \times 10^{-4}$, respectively, one-sided paired Wilcoxon test), but they pertain only to detected recombination breakpoints. As discussed previously, breakpoints are detectable only when the parental strains are distinctive enough that exchanged blocks carry within them distinctive patterns of bases, and therefore the detected recombination rate is an underestimate of the true recombination rate.

**Recombination hotspots**. One open question when studying recombination is whether breakpoints are uniformly distributed across the HIV genome or whether instead they cluster preferentially in certain regions (called hotspots) while leaving others relatively intact (cold spots). To address this, we calculated hot and cold spots of recombination across the 3′ half genomes of the 9 participants using a sliding window of 20 nucleotides, and found that hotspots were mostly clustered in Env, before or after the variable loops, whereas regions where two genes overlapped (*vif/tat* and *vpu/env*) carried several cold spots (Fig. 6).

**Increased genetic complexity among recombinants**. Across all nine participants, diversity within recombinant sequences was higher than intra-T/F lineage diversity but lower than inter-T/F diversity (Fig. 7). Previously, we observed that in homogeneous

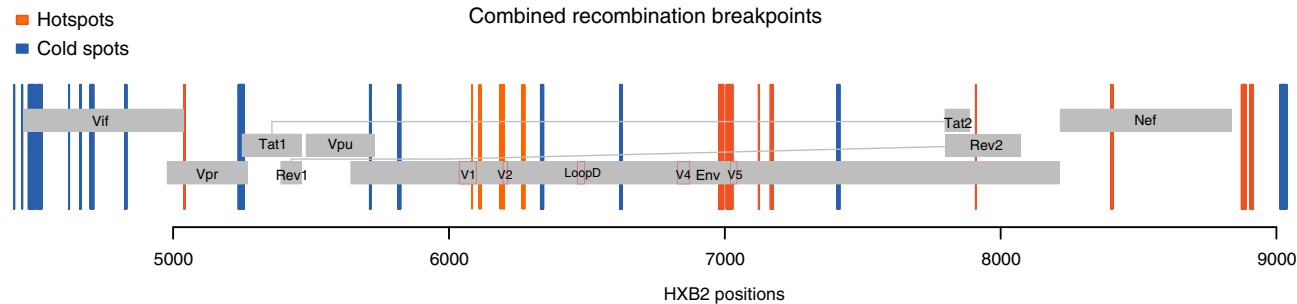

**Fig. 6** Identification of recombination hotpots in HV-1 genome. Recombination hotspots are shown in dark orange and cold spots in blue across the 3′ half genome. Position numbering is relative to HXB2. Each line represents a position, and the thickness of the colored regions represents consecutive positions. The sequences from the last time point (day 432) in CH0654 were excluded from the analysis due to the initiation of ART at day 112

infections recombination lowers the rate at which diversity accumulates in the viral population[35]. In these nine heterogeneous infections, recombination not only tends to lower the average diversity, but it also broadens the spectrum of genetic diversity distribution (Fig. 7). We speculate that this mechanism allows the virus to explore a wider range of phenotypes, and therefore potentially more likely escapes from immune responses. Similar patterns of recombination were observed in vitro (Supplementary Fig. 18).

**Recombination analysis in a homogeneous infection**. Because of its sensitivity in detecting recombination even in low-diversity scenarios, RAPR also proves useful when studying viral evolution by recombination in homogeneous infections, once enough diversity has accrued to enable its detection. We show one example of this application, and use it to address the question of whether recombination contributes to the emergence of resistance. We used viral and neutralization data from CH0505 who developed broadly neutralizing antibodies (bnAbs) over the course of several years, and whose antibody lineages and neutralization activity have been well characterized[30,36,37]. The first mutations conferring resistance arose at day 90 (Fig. 8a). New mutations continued to accumulate, and the ones conferring resistance persisted at later time points. Two CD4 binding site (CD4bs) targeting B-cell lineages were found in this individual, and by week 100 (day 692) into the infection, the virus developed complete resistance to the CH235 antibody lineage and to half of the antibodies from the CH103 lineage (Fig. 8b). As expected, given the homogeneity of a single T/F virus infection, RAPR does not find any evidence of recombination at the early time points. By day 538, however, it finds that 25 sequences out of 31 are recombinants, and by day 692, all sampled sequences are either novel recombinants or descendants of recombination events (Fig. 8c).

Interestingly, the mutations that confer CH505 autologous antibody resistance were maintained in the diversifying quasispecies through recombination events. For example, the pair of mutations V281G and N279D in the D loop appeared first at day 202 and day 55, respectively, and each mutation confers partial resistance to the CH235-lineage antibodies (Supplementary Table 4), with a strengthening of such resistance when they appear together[30,36]. RAPR finds that when parental strains are heterogeneous at position 279, the recombinant carries the resistant D genotype 10 out of 12 times. When the parental strains are heterogeneous at position 281, the recombinant carries the G-resistant form 9 out of 10 times. Seven out of 8 times both mutations are present in one of the parents, and both are inherited by the recombinant (Supplementary Fig. 19a). Therefore, whether at position 279, 281, or both, recombination carries the resistance mutations forward in the vast majority of cases

where the parental strains were discordant. The glycosylation site conferring mutation T234N was also found to be preferentially carried forward by recombination events in CH505. It was first detected at day 202 (Fig. 8 and Supplementary Table 4) and by the time there is enough overall diversity for RAPR to detect recombinants (day 538), most sequences carry the glycosylation site. There were only 6 recombinants from day 538 where the parental strains are heterogeneous at this position (i.e., only one of the parents carries the glycosylation site), and in all events the recombinant always carries the resistance-conferring genotype (Supplementary Fig. 19b), indicating that recombination contributes to the rate at which this mutation reached fixation. We hypothesized that the glycosylation site mutation at position 234 could confer resistance to autologous antibodies, and confirmed experimentally that it indeed confers resistance to early lineage CD4bs antibodies isolated from CH0505 (Supplementary Table 4). Overall, RAPR found 17 unique recombination events and 11 descendants, all in the last two time points from CH0505. The other tested tools (RAT[38], RDP4[24], RECCO[23], and GARD[27]) identified between 0 and 18 recombinants.

**Increased resistance of recombinants to neutralization**. CH0010 also developed potent neutralizing antibodies against all five T/F Env pseudoviruses over time (Supplementary Table 5). To determine if recombinants would allow viruses to escape from later developed autologous neutralizing antibodies, we looked at the sequences sampled at day 72 and generated Env pseudoviruses from two non-recombinants and three recombinants (Supplementary Fig. 20 and Supplementary Table 5). The two non-recombinant viruses could only be neutralized by the later time point plasmas (days 188 and 275). However, none of the three different recombinants from day 72 were neutralized by all plasmas, ignoring the weak neutralization of one day 72 Env pseudovirus by the plasma from day 188 (Supplementary Table 5). This indicated that recombinants were more likely be resistant to later autologous neutralizing antibodies than non-recombinants.

**Discussion**

We developed a new analytical tool, RAPR, that allows for computationally efficient detection of recombinants in longitudinal samples from recently infected individuals. RAPR classifies sequences depending on whether they originated from the T/Fs via mutation or from a recombination event between T/F lineages. Compared to existing programs like RDP4[24] and GARD[27], our simple simulations indicate that RAPR is more sensitive in detecting recombination events particularly in low-diversity settings (e.g., about 1%). RAPR has a built-in mechanism to constrain later sequences from being considered as parental strains of earlier sequences. RAPR sensitivity, detailed

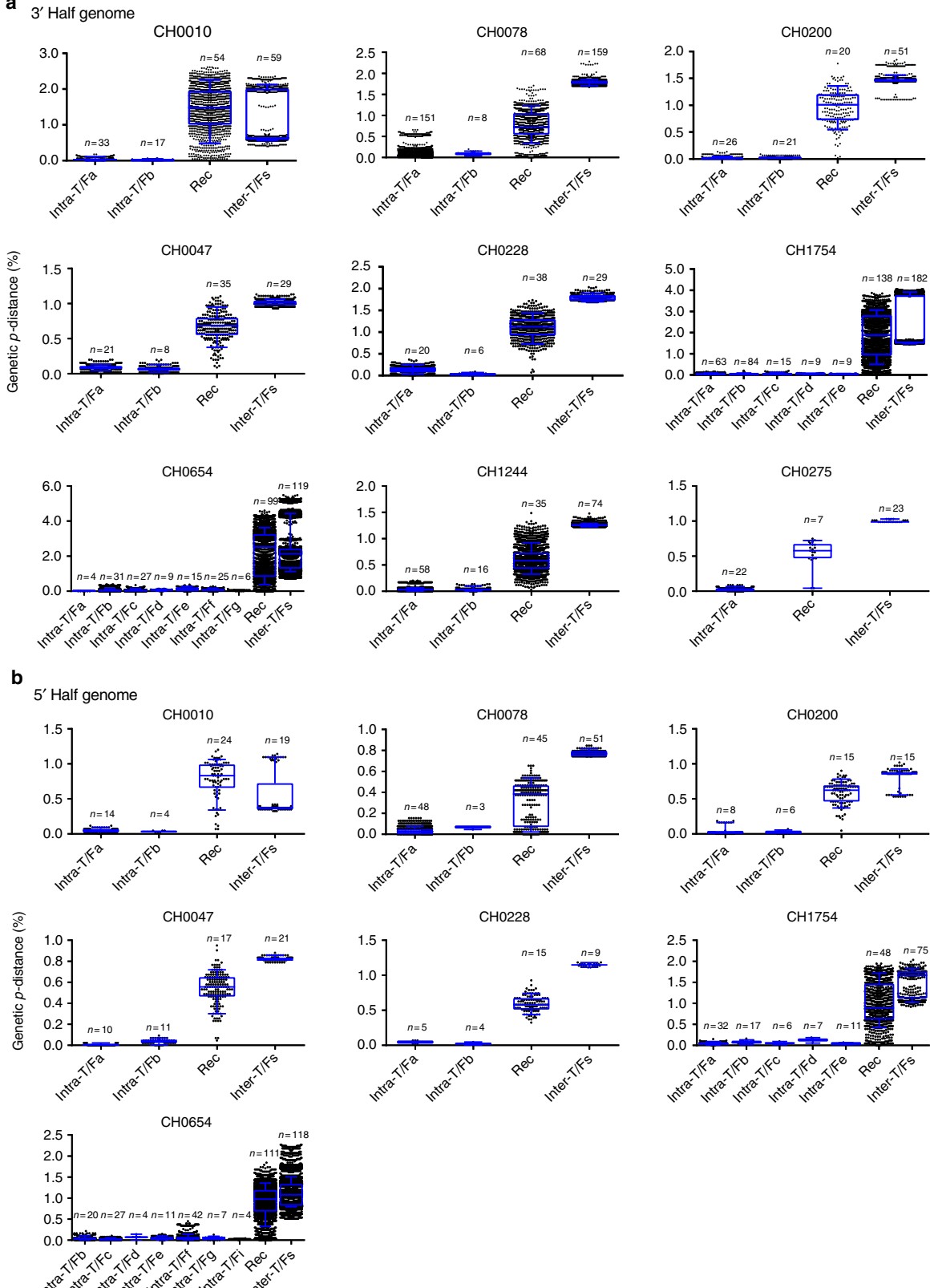

**Fig. 7** Genetic *p*-distances within lineages, inter-lineages, and within recombinants. Pairwise genetic distances (*p*-distance) were calculated among recombinant viruses, variants evolved from the same T/F virus (intra-T/F), as well as variants from different T/F viruses (inter-T/F) for 3′ half (**a**) and 5′ half (**b**) genomes. For intra-T/F diversity, T/F lineages with less than three sequences were excluded. Pairwise genetic distances are shown in black dots, and their means in blue lines. All data were calculated by combining the viral sequences from all time points up to the time that all of the viruses were recombined. The middle line indicates the median of the diversity; the box shows 25 to 75 percentile of the data; and the whisker shows 10 to 90 percentile of the data. The number of sequences for each pairwise comparison group is indicated at the top of the plot

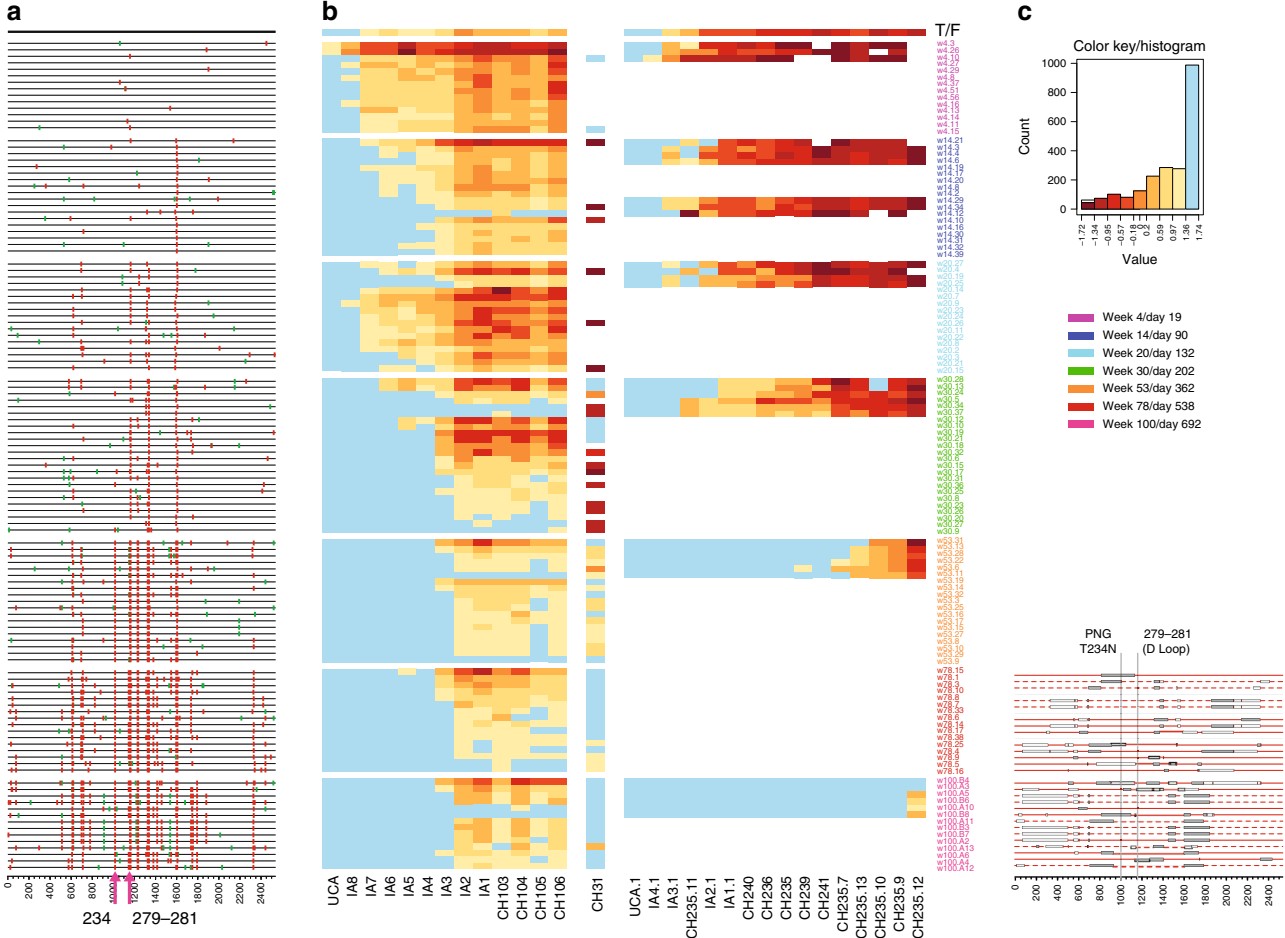

**Fig. 8** Sequence and neutralization analysis of viruses from homogeneously infected CH0505. **a** Syn/nonsyn highlighter plot shows the synonymous (green) and nonsynonymous (red) mutations compared to the T/F sequences. **b** Autologous neutralization analysis. Multiple Env pseudoviruses were generated from seven time points from CH0505 over 2 years of infection. Neutralization susceptibility of 123 pseudoviruses were determined with 13 CH103-lineage antibodies (right panel) and neutralization susceptibility of 33 pseudoviruses were determined were determined with 16 CH235-lineage antibodies (right panel). Neutralization magnitude is color coded by warm colors, with red being the strongest. Blue color indicates that no neutralization activities were detected with neutralizing antibodies at the 50 µg/ml concentration. **c** Recombinant *env* genes from days 538 and 692 after infection. Recombinants and their breakpoints, as determined by RAPR, are shown. Recombination breakpoints are shown as black boxes, and they are filled when the breakpoints are significant

output, and unique time-of-sampling features enabled an informed exploration of the timing of replacement of the T/F lineages by either recombinants or their descendants. This is important for understanding the continuous evolution of the viral population and the rapid exploration for new immunological escapes. RAPR thus nicely adds to an existing set of other tools for detecting and characterizing recombination in vivo that results from continuous interplay among distinct viruses during infection[39,40].

We applied RAPR to SGA-derived sequences sampled longitudinally from 9 heterogeneously infected participants, starting from very early in infection. We calculated frequencies of accumulation of new mutations and new breakpoints, as well as hot and cold spots for recombination. The accumulation rate of new mutations was higher than the accumulation rate of new breakpoints across all participants, although this may be due to the potential underestimation of recombination events in homologous regions. Even then, recombinants quickly replaced the T/F lineages with a median half-time of 27 days, and the recombination significantly increased the genetic complexity of a viral population. The rapid loss of T/F lineages should be taken into account if a reduction in numbers of T/F viruses is a parameter

used to evaluate the effectiveness of a prevention or vaccination strategy[41]. This is compatible with previous studies, which have found that by reassembling genotypes that have appeared through independent events, recombination can generate a wider range of new phenotypes that would either take much longer or be less likely to appear through mutation alone[6]. This wider spectrum of genetic variants generated by recombination might allow viruses to rapidly adapt to the host environment and outcompete parental viruses during infection.

We also showed how RAPR can be applied to longitudinal sequences sampled from homogeneous infections. Once adequate diversity has accumulated in the HIV-1 quasispecies, all triplet combinations of sequences can be tested, and recombination events framing resistance mutation can be tracked as they emerge over time. The data can then be used to answer the question of how recombination affects the establishment of key antibody-resistant mutations.

Recombination rates calculated in vitro tend to generally be higher than the ones observed in this study[18–20]. Bioinformatics recombination tools cannot detect recombination events between highly related parental strains within the same lineage or that occur in highly similar regions of the parental genome. To avoid

this limitation, Cromer et al.[31] developed a statistical method that accounts for template switching modeled on in vitro results. In our study, we used a simulation approach in which we introduced known numbers of breakpoints in simulated data to estimate strand switching events in vivo. Another caveat in the interpretation of our results is that as the sampling times increases and sequences become more divergent, the chance of a convergent selection event increases as well, thus dampening RAPR ability to distinguish independent recombination events. The new mutation and new breakpoint rates were found to peak at coordinated times (Fig. 5)—a phenomenon particularly striking in the 3′ half genomes. This suggests that there may be transient periods of stronger diversifying selective pressure acting on both recombination and base-substitution rates during the course of an infection.

The breakpoints detected by RAPR, provided in parallel to the phylogenetic trees, can help clarify lineage relationships. When combining 3′ half genome data from all nine participants, we found that new breakpoints tended to accumulate around (either before or after) the variable regions of Env (hotspots). In contrast, in vitro studies of inter-subtype and CRFs have shown that sequence homology drives recombination, with breakpoints clustering in more conserved regions between the parental strains[42,43]. More recently, Jia et al.[44] analyzed breakpoints in CRF sequences from the Los Alamos HIV Sequence Database and found hotspots to be clustered at protein overlapping regions, but not within the Env protein. The fact that we observe recombination hotspots in Env around variable regions during within-host evolution could be due to a combination of factors. For example, recombination events mechanistically favor regions of local homology of greater than 30 bases (maximum efficiency is seen at 60 bases or longer) to facilitate strand switching during reverse transcription[45], although strand transfers were frequently detected at limited homologous regions (<2 bases) in our recent study[46]. Between highly diverse viruses representing different subtypes, this may severely constrain the rate of recombination events in the variable env gene. In contrast, the degree of homology throughout the genome in our setting of highly related T/F variants is much greater, including homology in Env, enabling recombination throughout the genome. This, combined with the fact that Env is under a particularly high degree of selective pressure in vivo due to antibody-mediated selection for immune escape[47,48], could explain why we see an enrichment for recombination in variable regions in particular. In turn, the spectrum of genetic variants generated by recombination during infection contributes to the viral diversification that precedes the development of antibodies[40] with neutralization breadth during the course of an infection[36,37].

The extensive recombination observed in each of these nine participants highlights an important caveat on the interpretation of phylogenetic trees that has long been known[49], but generally underappreciated. Recombination violates the primary assumptions on which bifurcating evolutionary trees are built, yet such trees are frequently used to represent HIV-1 evolution within a host (including in this paper), and in the population as a whole. The trees provide a useful visualization of the clustering of related sequences and a portrait of the acquisition of diversity over time. However, in light of the extensive and early recombination demonstrated here, it is important to consider such HIV-1 trees as representations of clustering patterns of similar sequences, and not direct evolutionary trajectories.

Recombination plays a role in escape from T-cell responses[10,11] and bnAbs[50]. We now show that recombination between different T/F env genes in a heterogeneously infected individual were more resistant to the later autologous neutralizing antibodies than the T/F lineage variants from the same time point.

Recombination between different nAb escape mutations in a homogenously infected host likely rendered recombinants to more easily escape early maturation members of the bnAbs detected in the same individual. Taken together, these results suggest that extensive in vivo recombination may allow the complex recombinant population to have better chances to adjust to host immune selection pressure.

Finally we would like to note that intra-subtype recombination in HIV was first detected due to the high diversity of the virus, but it has also been observed in less diverse viral species such as hepatitis B viruses, enterovirus, and norovirus[51–53]. Because of its higher sensitivity in low-diversity scenarios, RAPR may prove useful in detecting recombination in other viral species where it has been more challenging to detect recombination.

## Methods

**Study participants**. To study HIV-1 recombination longitudinally, we identified recently infected participants whose estimated time from HIV-1 infection to blood sampling was generally less than 1 month. Of 36 individuals from a CHAVI001 infection cohort[33], 27 were infected with a single virus (homogeneous infection), and 9 were infected with multiple ones (heterogeneous infection). The first viral RNA-positive plasma samples from these 9 heterogeneous individuals were collected 9–27 days post infection, as estimated using our tool Poisson Fitter[54] (Table 1). This early into the infection, the within-lineage diversity was sufficiently low to allow us to infer infecting T/F lineages[33] and to estimate the time since infection using these previously described methods[54]. The number of T/F viruses we were able to identify in the 9 participants ranged from 2 to 9 (Fig. 1, Table 1, and Supplementary Figs. 1 and 2). Given the close similarity of the within-host T/Fs (the within-host T/F mean base pair distance ranged from 0.76% to 2.93%), we inferred that each participant had been infected by a single donor and at one single time point, with one notable exception: in CH0200 we noticed the appearance of a new, distinct lineage at day 74 that persisted at later time points. After excluding the possibility of sample contamination, we concluded that this was an incidence of superinfection. Near-full-length sequences of overlapping half genomes were obtained by SGA for 8 of 9 participants over multiple time points, while only the 3′ half genome sequences were obtained for CH1244.

Twelve homogenously infected and 7 heterogeneously infected individuals had no known protective or disease susceptible HLA alleles (Supplementary Table 6). In order to have enough diversity to detect recombination from early infection, as matter of necessity, we focused on heterogeneously infected participants. As a result, this study population may have biases relative to single T/F infections in terms of persistence of viral diversity. In addition to these 9 heterogeneous participants, we also included 341 env sequences from the homogeneously infected CH0505, spanning day 19 through day 692 since infection[36].

Longitudinal plasma samples within 2 years of infection were obtained from nine heterogeneously infected participants as well as one homogenously infected participant. All participants were male. CH0010, CH0047, CH0078, CH0200, CH0228, CH0275, CH1244, and CH1754 were from Malawi, CH0078 from South Africa, and CH0654 from the United States. None of these nine individuals developed AIDS-like symptoms within 2 years of infection. No individuals were on antiretroviral therapy (ART), except CH0654 who was on and off ART since day 112 after infection. Therefore, sequences from the last time point (day 432) of CH0654 was excluded from all analyses. The first screening plasma samples were collected within weeks from the infection for eight individuals, on or before Fiebig stage III, while the screening sample for CH1244 was collected at Fiebig stage IV[55]. All viruses were subtype C except CH0654, which was subtype B. Written informed consent was obtained from all study participants and the study was approved by the Duke University Institutional Review Board.

**Amplification of near-full-length viral genome by SGA**. Viral RNA was extracted from plasma samples or culture supernatant using the PureLink Viral RNA/DNA Mini Kit (Invitrogen, Carlsbad, CA). Complementary DNA was synthesized using the SuperScript III reverse transcriptase (Invitrogen, Carlsbad, CA) with the primers 07Rev8 (5′-CCTARTGGGATGTGTACTTCTGAACTT-3′; nt 5193–5219 in HXB2) for 5′ half genome SGA, or primer 1.R3.B3R 5′-ACTACTTGAAGCACTCAAGGCAAGCTTTATTG-3′ (nt 9611–9642) for the 3′ half or near-full-length genomes. SGA was performed to obtain the 5′ half, 3′ half, or near-full-length HIV-1 genome as described previously[56]. For the 5′ half genome amplification, the first round PCR was carried out using the primers 1.U5.B1F 5′-CCTTGAGTGCTTCAAGTAGTGTGTGCCCGTCTGT-3′ (nt 538–571) and 07Rev8 5′-CCTARTGGGATGTGTACTTCTGAACTT-3′ (nt 5193–5219), and the second round PCR with primers Upper1A 5′-AGTGGCGCCCGAACAGG-3′ (nt 634–650) and Rev11 5′-ATCATCACCTGCCATCTGTTTTCCAT-3′ (nt 5041–5066). To amplify the 3′ half genome, the first round PCR was performed using the primers 07For7 5′-CAAATTAYAAAAATTCAAAATTTTCGGGTT-TATTACAG-3′ (nt 4875–4912) and 2.R3.B6R 5′-TGAAGCACTCAAGG-CAAGCTTTATTGAGGC-3′ (nt 9636–9607), and the second round PCR with

primers VIF1 5′-GGGTTTATTACAGGGACAGCAGAG-3′ (nt 4900–4923) and Low2c 5′-TGAGGCTTAAGCAGTGGGTTCC-3′ (nt 9591–9612). Then, 2 μl of the first round PCR products were used for the second round PCR. The PCR thermocycling conditions were as follows: one cycle at 94 °C for 2 min; 35 cycles of a denaturing step at 94 °C for 15 s, an annealing step at 55 °C for 30 s, an extension step at 68 °C for 5 min; and one cycle of an additional extension at 68 °C for 10 min.

For amplification of near-full-length HIV-1 genome, the first round PCR was carried out using the primers Upper1A and 2.R3.B6R, and the second round PCR was done with the primers Upper2 5′-CTCTCTCGACGCAGGACTCGGCTT-3′ (nt 681–704) and LTRD 5′-CTGGAAAGTCCCCAGCGGAAAGTC-3′ (nt 9460–9437). Then, 2 μl of the first round PCR products were used for the second round PCR. The PCR thermocycling conditions were as follows: one cycle at 94 °C for 2 min; 10 cycles of a denaturing step at 94 °C for 15 s, an annealing step at 55 °C for 30 s, an extension step at 68 °C for 8 min; 20 cycles of a denaturing step at 94 °C for 15 s, an annealing step at 55 °C for 30 s, an extension step at 68 °C for 8 min with an incremental 20 s for each successive cycle; and one cycle of an additional extension at 68 °C for 10 min.

**Sequence analysis**. The PCR amplicons were directly sequenced by the cycle sequencing and dye terminator methods on an ABI 3730xl genetic analyzer (Applied Biosystems, Foster City, CA). Individual sequences were assembled and edited using Sequencher 4.7 (Gene Codes, Ann Arbor, MI). The sequences were aligned using CLUSTAL W, and the manual adjustment for optimal alignment was performed using Seaview. Neighbor-joining trees were constructed using the Kimura 2-parameter model with 1000 bootstrap replications. GenBank Access numbers for the newly obtained sequences in this study are: MF499156–MF502416. These 3260 new sequences augment preexisting data to provide a unique dataset, including extensive sets of longitudinally sampled sequences from individuals infected with multiple T/F viruses. All sequences from this study, alignments, and auxiliary data are also available in the HIV special interest alignments (https://www.hiv.lanl.gov/content/sequence/HIV/SI_alignments/datasets.html).

**Generation of infectious molecular clones**. The infectious molecular clones (IMCs) representing T/F viruses from CH0200 (a, b, and c) and CH0228 (a and b) were chemically synthesized and cloned in pUC57 as described previously[56]. The virus stocks were prepared from the supernatants of 293T cells (NIH AIDS Reagent Program, cat. number: 103) transfected with IMCs[57]. All cell lines were tested free of mycoplasma contamination.

**Virus culture**. Primary CD4$^+$ T cells were purified from peripheral blood mononuclear cells from a healthy donor under the protocols approved by the Duke University Institutional Review Board. Written informed consent was obtained from all study participants. Cryopreserved CD4$^+$ T cells were thawed and stimulated for 3 days in RPMI-1640 containing 10% fetal bovine serum (FBS), interleukin 2 (IL-2) (32 U/ml; Advanced Biotechnologies, Columbia, MD, cat. number: 03-001-050), soluble anti-CD3 (0.2 μg/ml; eBioscience, San Diego, CA, cat. number: 16-0037-81), and anti-CD28 (0.2 μg/ml; BD Bioscience, San Diego, CA, cat. number: 16-0281-81). The stimulated cells ($1 \times 10^6$ cells) was seeded into each well of a 96-well plate, and infected with a mixture of multiple T/F viruses (0.001 m.o.i. (multiplicity of infection) for each virus) from each participant (3 T/F viruses for CH0200, and 2 T/F viruses for CH0228)[57]. After absorption at 37 °C for 4 h, the cells were washed 3 times with RPMI-1640. The infected cells were cultured in a 24-well plate with 600 μl of RPMI-1640 containing 10% FBS and IL-2 (32 U/ml). The supernatant (200 μl) were harvested at day 4 and then passaged onto fresh CD4$^+$ T cells four times. The virus replication was monitored by determination of the p24 concentration in the supernatant using the p24 ELISA kit (PerkinElmer, Waltham, MA). The near-full-length genomes from passages 1, 2, and 4 were obtained by SGA to determine the frequency of recombination in vitro.

**Neutralization assay**. Neutralization activity was measured in a luciferase reporter system in TZM-bl cells (NIH AIDS Reagent Program, cat. number: 8129). Plasma samples were heat inactivated at 56 °C for 1 h and were then diluted in a 1:3 serial dilution starting at 1:20. The diluted plasma samples in duplicate were incubated with viruses for 1 h at 37 °C and then used to infect TZM-bl cells. The 50% inhibitory dose (ID$_{50}$) was defined as the plasma dilution at which relative luminescence units (RLUs) were reduced by 50% compared with RLUs in virus control wells after subtraction of background RLUs in cell control wells. A response was considered positive for neutralization if the ID$_{50}$ titer was >1:20 dilution.

**Determination of dates post infection**. Highlighter plots and neighbor-joining phylogenetic trees were generated using the Highlighter tool at the Los Alamos HIV sequence database (https://www.hiv.lanl.gov/content/sequence/HIGHLIGHT/highlighter_top.html). Based on visual inspection of these plots we determined that all nine participants in this study had been infected by two or more genetically distinct T/Fs. Days since infection were estimated by applying the Poisson fitter model[33,54] on the first time point alignments from both the 3′ half and 5′ half genome sequences and then taking the harmonic average between the two time

estimates. In order to account for the heterogeneity of the infection, we divided each first-time-point sequence alignment into single-founder lineages. Sequences significantly enriched for hypermutation ($p < 0.1$ by Fisher's exact test)[58] and sequences with ambiguous nucleotides were excluded from the analysis. Our method neglects differences between the putative ancestor and the consensus, an assumption that breaks down for very small sample sizes. For this reason, lineages with less than three sequences were excluded, and also because the goodness-of-fit test in the Poisson fitter is based on the chi square test, which is unreliable for small samples (less than 4 data points). For each remaining lineage, we approximated the T/F sequence by the lineage consensus, calculated the Hamming distance (HD) of each sequence in the lineage from this T/F sequence, and then merged all HDs from lineages of the same alignment into one distribution. Assuming that all T/Fs evolved from the same time point, were all equally fit, and accumulated random mutations at the same rate, the combined HDs follow a Poisson distribution. We proceeded to estimate the time since infection using this combined distribution following methods previously described[54]. In order to estimate the dates post infection for participants for which we had alignments covering both the 5′ half genome and the 3′ half genome, we first timed each half genome alignment separately, and then averaged the two estimates using a harmonic mean weighted by the inverse of the number of sequences in each genome half, a method that minimizes the asymptotic sampling variance.

**Wald–Wolfowitz Runs Test**. Recombination results in contiguous stretches of genetic information inherited, with random mutations, from each parent. Our program, RAPR, checks for patterns of inheritance in every set of three sequences in a given alignment. Specifically, RAPR identifies the sites that differ in exactly one of the sequences and applies the Wald–Wolfowitz Runs Test[28] to check if these sites cluster more than expected of random mutations. This is done as follows. Let A and B be the putative parental strains and C the recombinant one. Each position in C is either identical to A but not B, identical to B but not A, identical to both, or different to either one. In the two latter scenarios we say that the site is non-informative. Define an "A run" as the maximal non-empty segment of adjacent positions in C that are either non-informative or identical to A but not B. Likewise, a "B run" is the set of contiguous sites that are either non-informative or identical to B but not A. Let $m$ be the total number of positions where C is more similar to A; $n$ the total number of sites where it is more similar to B; and $r$ the total number of "A" or "B" runs. The probability that there are at most $r$ runs just by a chance ordering of independent mutations is given by:

$$P(R \leq r) = \sum_{k=2}^{r} f_R(k),$$

where the probability of a total of $k$ runs is given by:

$$f_R(k) = P(R = k) = \frac{\binom{m-1}{\lceil \frac{k-3}{2} \rceil}\binom{n-1}{\lfloor \frac{k-1}{2} \rfloor} + \binom{m-1}{\lfloor \frac{k-1}{2} \rfloor}\binom{n-1}{\lceil \frac{k-3}{2} \rceil}}{\binom{m+n}{m}}$$

with $a$ and $a$ standing for the least integer not less than and the greatest integer not greater than $a$, respectively (see Supplementary Fig. 21 as an example).

**Similarity clusters and correcting for circularity**. In cases where multiple viruses establish an infection, and likely transmitted founder strains (T/Fs) are resolved, T/Fs can be constrained to be parents in a recombination triplet, thus avoiding detection of recombination events that happened in the donor, prior to transmission. When the input alignment contains data from multiple time points, RAPR can assign likely parent–daughter relationships by factoring in the time of sampling. As a result, sequences from future time points do not affect the recombination inferences at earlier time points. When sampling is sparse, this constraint may be broken by the fact that the actual parental strains may have escaped sampling. In this case RAPR chooses the sequences that are closest to the actual parent even though they may have been sampled at a later time point. Multiple testing is addressed through a false discovery rate approach[59] applied to each time point individually. Using a greedy algorithm, sequences are grouped into closely related clusters such that differences within each cluster are likely to have arisen by mutation. Some clusters will have evolved directly from a TF. Others will represent recombination events that were indicated in the initial comparison of all sets of three sequences. For each recombinant cluster, RAPR calculates the minimum HD from the closest cluster, and compares it to the minimum number of mutations diverging from the parental strains. Based on which of the two numbers is lower, the cluster is "explained" as having arisen from another cluster by mutation, or as a de novo recombination of sequences belonging to other clusters. When a recombination event is called, the sequence in the cluster with the least number of mutations from the parental strains, the earliest parents, or with the lowest $p$ value is deemed the initial recombinant, and all other sequences in that cluster that share the same breakpoints are considered "descendants."

In the realistic case of multiple recombination events, or when one of the parental strains escapes sampling, a triplet will yield a low $p$ value when parents

and child swap roles. When such parent–daughter cycles are found, RAPR replaces the parents from the least likely configuration in the cycle with an alternate set, and, if no such choice is admissible, merges the current cluster to a similar one and deem the recombinants "descendants". For more details see our tool help page (https://www.hiv.lanl.gov/content/sequence/RAP2017/help.html).

**Breakpoint determination and frequency estimation.** For each recombinant, RAPR identifies intervals in which the breakpoints are most likely to have happened. Each breakpoint interval is examined and the run-test $p$ value is recalculated after removing the run next to it. If this new $p$ value is less significant than the overall $p$ value, we conclude that the breakpoint is contributing to the significance of the recombination. For each time point in the alignment, RAPR calculates two frequencies: the number of de novo breakpoints per sequence and the number of new mutations per sequence, where by "new mutation" we mean any nucleotide change that is not present in any of the T/Fs. Each mutation and breakpoint is counted only at the time it appears for the first time. These frequencies are then converted to rates by dividing by the sample time interval over which these mutations or breakpoints arose. Kendall correlations between these rates and VL were compared using a one-sided sign test. The VL was averaged between consecutive time points to account for the fact that mutations and recombination events occurred in the time period between the two consecutive time points when VL were sampled. We then calculated Kendall correlations $\tau$ between the breakpoint rate (calculated as described above) and VL, and separately between the breakpoint rate and the mutation rate. CH0275 was excluded from this analysis because of too few time points. For the rest of the participants, we observed that 6 out of 8 times the Kendall $\tau$ was higher for the correlation of the breakpoint rate with the mutation rate compared to that with the average VL, and the other two times they were equal. This was statistically significant ($p < 0.015$) when applying a one-sided sign test. For the 5′, we had only seven sets, and five of these showed the same trend, but in the other two, the breakpoint rate was more positively correlated with the VL. This was not statistically significant ($p > 0.23$).

**Breakpoint simulation.** In order to estimate the potential bias introduced by missing breakpoints that fall in regions of homology between parental strains, we devised the following simulation. First, to encompass the time when recombination happened, we chose participants for whom RAPR had not detected recombinants of recombinants at the first time points. These were the 5′ half from CH0047 and CH0200 (day 19 and 11 respectively), and the 3′ half from CH0228 (day 19) and CH1244 (days 12 and 15). To avoid overestimating the frequency of template switching, only sequences from the early time points of 4 participants (CH0047, CH0200, CH0228, and CH1244) were selected because they were sampled very early and had no evidence yet of recombination of recombinants. Cromer et al.[31] estimated 5 to 14 template switches per full genome, so for our half genome samples we considered a range of breakpoints from 1 to 8 and for each value we randomly generated 100 artificial recombinants using the sequences from the above participants. Each recombinant was generated randomly selecting two parents, one from each of two different T/F clusters. For each set of 100 recombinants, we counted how many breakpoints (significant and not) RAPR detected. We then accumulated, for each true breakpoint between 1 and 8, the total number of breakpoints and the number of statistically significant breakpoints RAPR detected (Supplementary Figs. 13 and 14).

We also used the same simulations to estimate frequentist 95% confidence interval on the true number of breakpoints as follows. For each recombinant observed in the four samples mentioned above, we randomly recombined the respective parents 5000 times, assigning breakpoints randomly drawn from a uniform distribution between 1 and 8. For every set of artificial recombinants from the same parental pair, we took all the recombinants for which RAPR detected the same amount of breakpoints as in the observed recombinant, and looked at the distribution of their true breakpoints (Supplementary Fig. 15). We then calculated the lower bound on the 95% CL on the true number of breakpoints. Similarly, we calculated the lower bound on the 95% CL on the true rate of recombination, i.e., the minimum rate of recombination necessary to explain the RAPR-observed recombinant. In particular, the lower level-$\alpha$ CL $l_\alpha$ on an estimate using statistic $m$ (for example, the observed number of breakpoints) for a population parameter $\mu$ (for example, the true number of breakpoints) is defined as a statistic such that $p_\mu(l_\alpha \leq \mu) \geq \alpha$, where $p_\mu$ is the probability distribution under the assumption that the population parameter is $\mu$ and the relation holds independent of $\mu$. To calculate such an $l_\alpha$ when the allowed values of $\mu$ are discrete, and the statistic $m$ is sufficient for $\mu$, we can proceed to simulate the sampling process for various proposals $\pi$ for the population parameter $\mu$, and obtain the fraction of samples for which the simulated estimate $s$ is greater than the observed estimate $m$, i.e., $f_\pi(s \geq m)$. Assuming the simulation is good enough to ignore the variance of $f_\pi(s)$, and this fraction is a non-decreasing function of $\pi$, the least value of $\pi$ at which this fraction becomes greater than $1-\alpha$ defines such an $l_\alpha$. We have no reason to believe that the RAPR estimate of the number of breakpoints is a sufficient statistic, so the confidence interval determined in this fashion is likely to be overly conservative.

Alternatively, if we assume a uniform rate $r$ of recombination events per sequence position, then the true number of recombination breakpoints $s$ in a sequence of length $L$ are expected to be distributed as a binomial with size $L-1$ and probability $r$. Then, the probability of observing $m$ or more breakpoints $p_r(b \geq m)$

can be estimated as $\Sigma_\beta f_\beta(b \geq m) \, p_r(\beta)$. We can use this to find simulated CL on the rate of recombination: in particular, the least value of $r$ for which $p_r(b \geq m)$ becomes greater than $1-\alpha$ provides a CL for the rate.

All simulations were run using R[60]. The simulated recombinants, together with the parental sequences used to generate them, are available at the following link: (https://www.hiv.lanl.gov/repository/hivdb/Simulated_Data_for_RAPR_NatComm_Paper). RAPR is available for public use on the LANL HIV database (www.hiv.lanl.gov/content/sequence/RAP2017/rap.html). All software is developed in C++, R, and cgi. The R code utilizes functions from the packages ape[61] and seqinr[62].

**Neutralization analysis of viruses from CH0505.** To determine the impact of recombination on the emergence of antibody resistance, we ran RAPR on 341 env sequences from CH0505[36], spanning day 19 through day 692 since infection. Of these, 123 of these env genes were used to generate pseudoviruses which were tested against a panel of 13 autologous CH103-lineage antibodies, and 33 of the 123 were additionally tested against a panel of 16 autologous CH235-lineage antibodies.

**Identification of recombination hotspots.** To explore possible hotspots of recombination across the 3′ half genomes, we devised the following strategy. For every participant, we counted the appearance of a new breakpoint only once. Because the program outputs an interval where the recombination breakpoint is most likely to have occurred, we took the midpoint of such interval, and then considered the cumulative distribution of the positions at which these breakpoints occurred across all nine participants. Regions where the cumulative distribution had a steep step upward indicated a high accumulation of breakpoints and therefore hotspots. Conversely, regions where the cumulative distribution was flat indicated potential cold spots. To assess this with statistical significance, we considered a sliding window of 20 nucleotides and for each window we calculated the slope of the line that best fitted the cumulative distribution within said window. If there were hotspots of recombination within the given window, we expected the slope of the best fitting line to rise sharply, or to lower in areas of cold spots. We initially explored various window sizes, between 10 and 50 nucleotides, and we saw that decreasing to smaller sizes from 20 nucleotides did not add any additional information, while increasing the size caused several potentially interesting regions to go undetected. To smoothen out the differences across participants, we repeated the sliding window algorithm 9 times, each time removing one of the participants. For each iteration we considered all positions corresponding to the upper quartile of the slopes and all positions corresponding to the lower quartile, and then took all the positions that fell either in the upper or the lower quartile every time we ran the algorithm.

**Genetic distances.** Genetic complexity among lineages was measured as the mean pairwise $p$-distances. The $p$-distances (the proportion of sites at which two sequences differ) were calculated among recombinant viruses, variants evolved from the same T/F virus (intra-TFs), as well as variants from different T/F viruses (inter-TFs) for 3′ half and 5′ half genomes using the software MEGA6[63].

**Dynamical modeling.** We extended the basic mathematical model for HIV dynamics[64] to include the coinfection of target cells by two different viruses and the resulting generation of recombinant viruses. Assuming $n$ T/Fs, let $V_i$ be the density of the $i$th ($i = 1 \dots n$) T/F virus, and $V_{n+1}$ the density of recombinant viruses. Uninfected target cells $T$ become infected with at a rate $\beta'_i$, and produce virus at a rate $p_i$, for $i = 1 \dots n+1$. Free virus is cleared at a rate $c$. Cells infected with the T/F viruses can be coinfected with other viral strains, and such coinfection leads to the generation of recombinant viruses. It is well known that early after infection, the amount of CD4, the main receptor for HIV binding, on the surface of infected cells is downregulated due to the action of the Nef and/or Vpu proteins[65]. Although the kinetics of CD4 downregulation in vivo are not known, in our model we assume that downregulation is slow and incomplete, and that a cell infected with one viral variant can be reinfected with another viral strain. Coinfection occurs at a reduced rate $\gamma\beta'_i$. Uninfected and virus-infected cells die at rates $d$ and $\delta_i$, respectively. The dynamics of $n$ T/F viruses and a recombinant strain during acute HIV-1 infection is then described by the system of differential equations

$$\frac{dT}{dt} = d(T_0 - T) - T \sum_{i=1}^{n+1} \beta'_i V_i,$$

$$\frac{dI_i}{dt} = \beta'_i V_i T - \delta_i I_i - \gamma I_i \sum_{j=1}^{n+1} \theta_{ij} \beta_j V_j, \qquad i = 1..n,$$

$$\frac{dI_{n+1}}{dt} = \beta'_{n+1} V_{n+1} T - \delta_{n+1} I_{n+1} + \gamma \sum_{i=1}^{n} \sum_{j=1}^{n+1} \theta_{ij} \beta_j I_i V_j,$$

$$\frac{dV_i}{dt} = p_i I_i - c V_i,$$

where $I_i$ and $I_{n+1}$ is the density of cells infected with the T/F viruses and the recombinant virus, respectively, and $\theta_{ij} = 1$ if $i \neq j$ and $\theta_{ij} = 0$ otherwise. It should be noted that in this simple model we assumed that coinfection with two different viruses leads to a generation of the recombinant virus.

Previous studies have indicated that the virion clearance rate $c$ ranges from 9 to 36 per day[66] which is much faster than the estimated lifespan of infected cells ($\delta_i = 1$ per day)[67]. Therefore, given fast dynamics of the virus, the density of a given viral variant is approximately proportional to the density of infected cells producing this variant, $V_i = \frac{p_i}{c} I_i$, which simplifies the model. By replacing $\beta_i = \frac{\beta'_i p_i}{c}$ and $r_i = \beta_i T - \delta_i$, and denoting the frequency of the $i$th variant as $f_i = \frac{I_i}{I}$ and $I = \sum_{i=1}^{n+1} I_i$ as the total number of infected cells we obtain the simplified model

$$\frac{df_i}{dt} = f_i\left(r_i - \sum_{j=1}^{n+1} r_j f_j\right) - \gamma\beta I f_i (1 - f_i), \quad i = 1\ldots n,$$

$$\frac{df_{n+1}}{dt} = f_{n+1}\left(r_{n+1} - \sum_{j=1}^{n+1} r_j f_j\right) + \gamma\beta I \sum_{i=1}^{n} \sum_{j=1}^{n+1} \theta_{ij} f_i f_j,$$

$$\frac{dI}{dt} = I \sum_{i=1}^{n+1} r_i f_i.$$

To further simplify the analysis, we assume the infectivity of all variants to be the same ($\beta_i = \beta = $ const), while allowing for potential differences in the rate of virus production $_{pi}$ from cells infected by variant $V_i$.

The dynamics of the recombinant viruses in the population are determined by two processes: accumulation due to selective advantage ($s = r_{n+1} - \bar{r}$ where $\bar{r} = \sum_{j=1}^{n} r_j f_j / \sum_{j=1}^{n} f_j$) and accumulation due to high degree of cell coinfection with two different viruses ($\gamma\beta I$). The rate of coinfection of cells with 2 different viruses is approximately given by $F_c = \frac{\gamma\beta I}{\gamma\beta I + \beta T}$ (derivation not shown). To estimate the rate of coinfection of cells with different viruses we assume that the density of target cells remains constant after peak infection, and the total number of infected cells is constant. Extensions of the model relaxing this assumption will be presented elsewhere. We fit the model predicting the dynamics of the frequency of the recombinant viruses in each individual to the experimentally measured frequency, where the initial frequency of the different viral variants is taken directly from the data. Alternatively, we assumed that recombinants were present at some frequency and their accumulation was due to selective advantage $s$ which was estimated by fitting the model (reduced to a logistic equation) to data. The model is fitted using maximum likelihood method as was recently described elsewhere[68]. Confidence intervals for estimated parameters were calculated by resampling the data using Jefferey's intervals for binomial proportions[69] with 1000 simulation runs. Because of the limited data, two parameters (the coinfection rate and selection coefficient) could not be accurately estimated from the data. Therefore, in a set of analyses we fixed the selection coefficient to various values and estimated the coinfection rate. Because of two half genomes available for some of the time points, for those individuals we have two independent measurements of the frequency of different viral variants, and these provide two estimates of the coinfection rate based on the dynamics of recombination at the 3′ and 5′ half genomes. Consistency of the estimated coinfection rates varied by patient. In some cases (e.g., CH0654, CH0200), we found nearly identical estimates for coinfection rates using 3′ and 5′ data (Table 2 and Supplementary Table 3). However, in other instances, estimates for predicted coinfection rate were different between 3′ and 5′ half data (e.g., CH0228, CH0078; Table 2 and Supplementary Table 3). Underlying reasons for such difference are not entirely clear but may be related to the different numbers of sequences analyzed. All models were implemented and run in Mathematica[70] (ver. 11.2).

**Data availability**. All newly generated nucleic acid sequences in the current study were deposited in GenBank with accession numbers MF499156–MF502416. RAPR is available for public use on the LANL HIV database (http://www.hiv.lanl.gov/content/sequence/RAP2017/rap.html). The simulated recombinants are available at the following link: (https://www.hiv.lanl.gov/repository/hivdb/Simulated_Data_for_RAPR_NatComm_Paper). All relevant data are available upon request.

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

## Acknowledgements

We thank Yi Yang and Sheri Chen for technical assistance; and Jennifer Kirchherr and Caroline Cockrell for compilation of clinic data. This work was supported by NIH grant NIH R01AI087520, NIH grants to the Center for HIV/AIDS Vaccine Immunology (AI067854), the Center for HIV/AIDS Vaccine Immunology and Immunogen Discovery (AI100645), the HIV/SIV Database and Analysis Unit (AAI 12007-0000-01000), the American Heart Foundation grant to V.V.G., and with Federal funds from the National Cancer Institute, National Institutes of Health, under Contract No. HHSN261200800001E. The content of this publication does not necessarily reflect the views or policies of the Department of Health and Human Services, nor does mention of trade names, commercial products, or organizations imply endorsement by the US Government.

## Author contributions

F.G., B.K., T.B., H.S., and E.E.G. conceived and designed the study. H.S., F.C., B.H., C.J., X.L., S.W., H.L., J.F.S., M.G.S., N.G., and B.F.K. performed experiments. E.E.G., T.B., O. C., G.A., H.Y., E.R., and B.K. developed the computational algorithm for recombination analysis. V.V.G. developed the mathematical model for recombination and viral load dynamics analysis. M.S.C. directed the clinical trial and supplied the clinical samples. H. S., E.E.G., V.V.G., F.C., B.H., P.H., X.L., J.F.S., N.G., B.F.K., D.C.M., M.S.C., G.M.S., B.H. H., A.J.M., B.F.H., T.B., B.K., and F.G. analyzed the data. H.S., E.E.G., V.V.G., P.H., N.G., A.J.M., G.M.S., B.F.H., T.B., B.K., and F.G. wrote and edited the manuscript.

## Additional information

**Competing interests:** The authors declare no competing interests.

