## [Peer Review File · Nature Communications]

Reviewers' comments:

Reviewer #1 (Remarks to the Author):

NCOMMS-17-19294

Title: Tracking HIV-1 recombination to resolve its contribution to HIV-1 evolution by a novel RAP tool

Overview: This is well written paper describing the development and use of a new tool (RAP—Recombination Analysis Program) in the setting of HIV. This new program aims to characterize how HIV recombines within a host during natural infection. This review with focus on areas listed below.

RAP. I think the analytical underpinnings of RAP are sound and will be very useful in the field. My main concerns with the program itself are how the validations were conducted; see below. I also think the details on how RAP was compared to RAT, GARD and RECCO are lacking to really understand claims of supremacy and sensitivity. It is also not clear how the authors determine that the RAP tool is 'reliable', as stated in the Discussion. For example, what was the gold standard used to compare against? I also do not understand the definitions of "hot" vs "cold" spots for recombination. The simulations are interesting but do not provide enough detail to understand how some were classified as hot and others cold. Recombination rate. I think the calculations of recombination rate are correct. I take some concern with the use of the term "highly significant". I also think that the authors are correct that the presented rate is an under-estimate but it should be qualified as "probably" and under-estimate because you cannot measure the 'true' rate to be certain.

Study Population. The authors have accessed a group of well-characterized individuals, which have an enormous amount of previously generated data. These data have been published in many publications, and most of the data presented in this paper have been published before. The authors use these data to test their RAP method. My issue with the study population is that there are only nine persons, and one of the CH0275 should be excluded because there are too few timepoints sampled and CH0654 should be excluded because that person used HAART during the sampling and was also a different HIV-1 subtype. Further, all of the participants are male and there is no description of route of infection, which may bias the conclusions of T/F strains. It is also not clear if the "heterogeneous" group that is being studied here meets the criteria for co-infection. Specifically, did the mixture of viruses observed transmitted from the same donor (mono-infection) or different donors (co-infection)? Further, the authors state that all participants were characterized during "acute" infection, but that is not true. Fiebig Stage III and after is really "recent", not "acute", as measurable immune responses to the infection can be detected during this time. In the Methods, the authors state that the 'days since infection' were based on a Poisson filter model. I do not quite understand why they did this. It seems that each of the participants had their days since infection estimated based on Fiebig staging, which is pretty standard in the field. Did the authors compare the sequence TMRCA analysis and Fiebig staging analysis? Also, for Table 1, I do not understand how people diagnosed in Fiebig III can have longer 'days of infection' than people in Fiebig IV?

T/F. Transmission/Founder (T/F) viruses have been a convenient way to understand and explain the beginning of HIV infection; however, there remains controversy whether or not the methods used to determine whether a particular HIV sequence represents a true T/F

variant is correct. The authors state "unambiguously" identify, but I do not think it is true. As noted in this paper's Discussion, the rapid loss of T/F variants provide doubt to me that we can truly say which variants represent T/F variants anyway. In particular, it is not clear to me (and others) that the analysis methods used to characterize HIV viral populations that have already adapted to the new host, especially in the setting of burgeoning immune responses, can adequately characterize T/F strains. This is especially pertinent in this study because: 1) the whole premise of this study is that the T/F strains are true representations of the starting viral populations, 2) they only looked at people with heterogeneous viral populations (although I do not quite understand what the cut-offs were for deciding how diverse a viral population had to be to be eligible for the study), and 3) there was quite a few days between infection and sampling so adaptation should have at least taken place to some extent, even better engagement with CD4 and CCR5. To this last point, I think the authors should use their tool to look at viral dynamics in monkeys who underwent dual infection with two known T/F to see if they get the same answers as they see in humans. I also do not understand why lineages with <3 sequences were excluded from the analysis. Further, since this paper only evaluates people with heterogeneous T/F populations, then there may be a selection bias for these people. What is the reason for such diversity that persisted? Was diversifying or purifying selection present; if so how was it measured? What if they applied RAP to the sequences generated from people with homogeneous viral populations? The hypothesis would be that no recombination would be detected, and I think that would be true, but maybe the investigators could check.

Immune selection and Viral load. The biggest weakness of this paper is the lack of measurement of immune responses (CTL, NAb, NK) to explain the observations presented. There is a very likely a selection pressure to drive recombination, or at least that hypothesis should be tested. The authors state this is likely but there is no follow up. Further, the authors discuss that immune pressures change (which is true) and are transient (which is usually not true). There should also be some evaluation for linkage of mutations with recombination. For example, if an escape mutation in one variant recombines with an escape mutation in another variant. I would also expect some evaluation of at least in vitro replication capacity for the recombinants to better understand why the virus recombines so much. The authors discuss HLA haplotypes broadly in terms of viral loads and disease progression (protective and susceptible), but it is not clear how it relates to recombination, and it really cannot be assessed without actually looking for escape mutations. (For example, even neutral or susceptible HLA induce CTL that can cause escape mutations.) There is some discussion about Nef escape mutations and maybe some escape mutations in Env from NAb, but how these escape mutations were determined is not clear from the paper. In particular, the determination of NAb breadth tells about escape of the whole virus but it does not tell us about which mutations were responsible for this escape and how these mutations influence the recombination observed.

The observations about VL are also interesting and how it may play in recombination. It is very clear from the literature that VL is directly related to CTL immune responses. It is also evident that VL and immune pressure are both related to recombination but the mechanisms and contribution of each are not assessed, since the effects of both (VL and immune response) are not independent of each other. There is some discussion or

protective and susceptible HLA types in relation to VL and viral diversity, which has been documented before, but the study does not clearly outline how this is relevant to recombination.

Reviewer #2 (Remarks to the Author):

The paper describes the development a new computer algorithm to genome recombination and its application to discern within patient (intra-quasispecies) recombination events during early phase HIV infection.

Recombination is one of the important mechanisms whereby HIV (and many other viruses) generate diversity that can impact on tropism and pathogenesis, immune escape and drug resistance. Studying recombination between highly diverse viruses (e.g. inter-subtype) is a relatively easy task. However, study of the virus population within an individual patient is complicated by the fact that the viruses are highly related, and breakpoint identification is difficult. The current method overcomes this by utilising runs testing to determine the randomness (or otherwise) of informative sequence states and from this the likelihood and location of recombination events. Applying this test to identify genetic recombination is not new, but its utilisation within a computational framework is, and it is this latter aspect that will render the analyses more applicable to large datasets and make it more accessible to the wider research community. Indeed its application for analysis of early stage HIV infection provides (subject to the caveats below) important insight into the role of recombination in early stage diversification of HIV and its possible impacts.

Points for clarification:

Line 280 – “RAP also assumes that parental 281 sequences are not sampled later than the recombinant”. Is this a fair assumption - surely, the whole point about viruses that from proviral reservoirs is that parental viruses can re-emerge? Given that it is impossible to sample the entire reservoir at any given time, it is feasible for viruses not sampled in the initial time points to ‘reappear’ later. This needs very careful consideration.

Lines 370-388 and 390-415. These sections need toning down!! Whilst the authors show that sequence variants across key nef- CTL epitopes show plasticity through recombinant, they show no direct evidence that this has been driven by CTL escape or indeed allows this virus to escape those responses in this particular patient. Similarly the narrative around neutralisation also relies on surrogate markers of neutralisation and lacks appropriate necessary direct biological data. Without associated phenotyping data on neutralisation or CTL escape, these statements are all hypothesis and supposition. Intriguing perhaps, but they need validating.

Reviewer #3 (Remarks to the Author):

Summary

This paper has two goals, one is an empirical paper tracking HIV recombination and a second goal describing a new tool (software) to characterize recombination. Unfortunately, the two goals get very confounded in this paper. On the second goal, the paper fails for a number of reasons. First, to introduce new software in an analysis setting like this, it is customary to evaluate the software, ideally with simulated data to show it is performing as expected. The authors do this to a certain extent, but they seem to do it as an after thought. In fact, the methods are actually described in the results section. They conclude from this effort that their approach underestimates recombination, but do not report on type I or type II error rates. Second, it is also appropriate to compare the new tool to existing tools to see if there is, indeed, better, worse, or similar performance. There are no data presented in the paper that give the reader any confidence that this new tool does anything useful. The authors do compare their approach to a few other methods, but using the empirical data. They all give different results. Because the data are empirical, you don't know the truth (the true frequency of recombination, the actual parents, or the breakpoints). So all you can conclude is that different methods give different results. Importantly (and somewhat disturbingly), there is already a very heavily used tool called Recombination Detection Program (RDP) – compared to the authors' 'Recombination Analysis Program (RAP). The RDP program is now in its fourth version and across all four versions the software has over 3500 citations. It is very surprising that the RDP package is not even mentioned nor cited in this paper. The fact that the authors have such a very similar name is concerning. Either way, this paper cannot move forward without a direct comparison to RDP.

While the authors set up an interesting experiment with some wonderful data, the results are still ambiguous because they do not actually know the true number of recombination events nor the breakpoints. So they cannot accurately calculate type I and type II error rates in RAP. The reader is then left with yet another method that claims to detect recombination. While the authors go through significant effort to generate simulated data, it is underutilized in terms of testing the method. They should report false positive rates and false negative rates and compare to other approaches (especially, RDP).

In the end, the manuscript is exceptionally long and cumbersome. It has methods in the results and goes back and forth between the biology and the recombination detection method. The work would benefit significantly from a partitioning of the software as its own paper with appropriate validation and comparison. Then an empirical paper on the application to this interesting HIV data set.

Other Issues

Pg 4, line 83 'between different studies' should be 'among'

Pg. 5, line 111 'SplitsTree provides a network' is one of many software packages that estimate and/or visualize gene genealogies as networks.

Pg. 5, line 116 'Defining the frequency of recombinants is particularly important in phylogenetic ...' Identifying the endpoints is also important.

Pg 6, lines 118-122 – GARD does not provide a list of recombinants, but see RDP! This

software implements a number of different approaches to detecting recombination, including a runs test (GENECONV). <http://web.cbio.uct.ac.za/~darren/rdp.html>

So contrary to the statement on page 6 that the runs test has not been implemented, it in fact has in RDP. This is the reference for the runs test applied to recombination: Padidam, M., Sawyer, S. & Fauquet, C. M. (1999). Possible emergence of new geminiviruses by frequent recombination. *Virology* 265, 218-225.

Page 10-11 lines 231-238, all this looks like methods to me, not results.

Page 16, line 356 sliding window size of 20. How is this justified?

Pg. 23, lines 520-521, how is homogenous versus heterogenous infection determined? Are there viral load, CD4+, etc. information on these individuals?

Responses to reviewers' comments

Summary

We thank the reviewers for their comments and have tried to address them all in full. In addition, we have made three major changes in the manuscript.

First, following the third reviewer's suggestion, **we have changed the software's name from RAP to RAPR.**

Second, **we now include systematic comparisons of our program to existing recombination detection tools**, demonstrating its greater sensitivity on simulated data, especially in the low-diversity setting it was designed for. We also highlight the features in RAPR that allow a deeper exploration of the implications of recombination *in vivo* compared to earlier tools.

Finally, **we have tracked the role of recombination in the emergence of antibody escape over time within a single subject, CH505**, for whom the antibody/viral co-evolution over time has been characterized in great detail, but for whom the role of recombination had not yet been explored. To include this new section, given space constraints, we removed an earlier section on T cell escape; we think the new example of antibody escape is more compelling and easier to follow. In addition, the inclusion of this new section enabled us to illustrate how to apply RAPR in a single T/F virus setting.

Below we address one by one the reviewers' comments.

Reviewer #1 (Remarks to the Author):

NCOMMS-17-19294

Title: Tracking HIV-1 recombination to resolve its contribution to HIV-1 evolution by a novel RAP tool

Overview: This is well written paper describing the development and use of a new tool (RAP—Recombination Analysis Program) in the setting of HIV. This new program aims to characterize how HIV recombines within a host during natural infection. This review with focus on areas listed below.

RAP. I think the analytical underpinnings of RAP are sound and will be very useful in the field. My main concerns with the program itself are how the validations were conducted; see below. I also think the details on how RAP was compared to RAT, GARD and RECCO are lacking to really understand claims of supremacy and sensitivity. It is also not clear how the authors determine that the RAP tool is 'reliable', as stated in the Discussion. For example, what was the gold standard used to compare against?

Answer: We thank the reviewer for the comments. We added more simulations where we created artificial recombinants from parental strains with different degrees of diversity between them. We now show that especially in low diversity settings, RAPR is able to detect far more

recombinant sequences than many of the other existing tools, including the widely used RDP4, currently the “gold standard” for recombination detection. Supplementary Table 2 shows how many known recombinants RAPR was able to detect in these low diversity simulated sets compared to RDP4 and the other tools included in the suite.

I also do not understand the definitions of “hot” vs “cold” spots for recombination. The simulations are interesting but do not provide enough detail to understand how some were classified as hot and others cold.

Answer: We now clarify in the main text the definition of hot and cold spots with the following sentence: “One open question when studying recombination is whether breakpoints are uniformly distributed across the HIV genome or whether instead they cluster preferentially in certain regions (called hotspots) while leaving others relatively intact (cold spots).” To explain more how hot/cold spots were determined, we also added in the Methods the following sentence: “Regions where the cumulative distribution had a steep step upward indicated a high accumulation of breakpoints and therefore hotspots. Conversely, regions where the cumulative distribution was flat indicated potential cold spots”. The work described in this particular section does not involve simulations, but rather analyses that allowed us to identify regions in the *env* gene where observed breakpoints in the real data tend to happen either significantly more often (hotspots) or significantly less often (cold spots) than in other regions of the genome across the 9 subjects presented in our study.

Recombination rate. I think the calculations of recombination rate are correct. I take some concern with the use of the term “highly significant”. I also think that the authors are correct that the presented rate is an under-estimate but it should be qualified as “probably” and under-estimate because you cannot measure the ‘true’ rate to be certain.

Answer: We have removed the adverb “highly” and thank the reviewer for the suggestion. As for the under-estimation, we show through our simulations that this kind of systematic bias is intrinsic to any recombination-detection method and it is indeed an under-estimate. We have now added Supplementary Figure 10 to better explain this phenomenon.

Study Population. The authors have accessed a group of well-characterized individuals, which have an enormous amount of previously generated data. These data have been published in many publications, and most of the data presented in this paper have been published before. The authors use these data to test their RAP method. My issue with the study population is that there are only nine persons, and one of the CH0275 should be excluded because there are too few timepoints sampled and CH0654 should be excluded because that person used HAART during the sampling and was also a different HIV-1 subtype. Further, all of the participants are male and there is no description of route of infection, which may bias the conclusions of T/F strains.

Answer: Longitudinal SGA sequences that span the full genome as overlapping half-genomes from individuals infected with multiple T/F viruses, as the ones described in this study, are only rarely reported. Except for CH0078, none of the sequences following the screening time point have been published before. We decided to include all available heterogeneously infected samples to be inclusive, which also allows us to increase the power of our statistical analyses.

The fact that subject CH0275 had only two time-points actually made it minimally affect our results, and we have already discussed the limitation carefully in the text. Therefore, we prefer to keep the subject while noting the sparse sampling.

As for CH0654, this subject was on and off ART since day 112 after infection, as described in the Methods. Therefore, we took out the data after day 112 for the VL dynamics. Because all viruses in CH0654 were recombinant in both genome halves at day 84, before the initiation of ART, the calculation of co-infection rate and the $T_{1/2}$ for recombinants to replace the T/F viruses was not affected by the treatment status.

It is also not clear if the “heterogeneous” group that is being studied here meets the criteria for co-infection. Specifically, did the mixture of viruses observed transmitted from the same donor (mono-infection) or different donors (co-infection)?

Answer: Thank you for the remark. We now clarify that, given our sampling, we are able to infer that each subject was most likely infected by a single donor. Of the 9 heterogeneously infected subjects, the mean diversity across different T/F lineages within a subject was 0.96% and 1.56% for 5’ half and 3’ half genome sequences, respectively, while it was 9.8% and 14.7% across subjects. Had these T/Fs not originated in the same donors, we would have observed comparable diversity within and across subjects. We did observe one likely incident of super-infection in subject CH0200, where a new lineage appeared at day 74 that persisted (in the form of a recombinant) to later time points. Even in this case, the super-infected virus was still from the same donor because the genetic differences between primary and super-infected viruses were too small to be from a different donor.

While the reviewer makes a valid point, the inference of the T/Fs is not a cardinal point in our analysis, rather, these sequences are treated as reference and therefore the program treats them as “parents” rather than recombinants. Equivalently, we could have run the same analysis without inferring T/F viral sequences and let the program assume that all sequences are potential recombinants. This could have potentially identified any of the T/F as a possible recombinant, but such recombination would have happened in the donor, not in the study subject. In order to focus on recombination events that happened in the study population only, we chose to label T/Fs and treat them as parental strains.

Further, the authors state that all participants were characterized during “acute” infection, but that is not true. Fiebig Stage III and after is really “recent”, not “acute”, as measurable immune responses to the infection can be detected during this time. In the Methods, the authors state that the ‘days since infection’ were based on a Poisson filter model. I do not quite understand why they did this. It seems that each of the participants had their days since infection estimated based on Fiebig staging, which is pretty standard in the field. Did the authors compare the sequence TMRCA analysis and Fiebig staging analysis? Also, for Table 1, I do not understand how people diagnosed in Fiebig III can have longer ‘days of infection’ than people in Fiebig IV?

Answer: We thank the reviewer for pointing out the correct terminology and we have indeed changed our language from “acute” to “recent” where appropriate. As discussed in detail in our previous publication (Lee et al., J Theor Biol. 2009), Fiebig stages define time windows rather than

definitive brackets, and when calculated cumulatively, the various stages do overlap in their 95% confidence intervals as shown by the following table (from Lee et al., J Theor Biol. 2009):

Stage	Duration of each phase (days)	Cumulative duration (days)
Eclipse	10 (7,21)	10 (7,21)
I (vRNA+)	7 (5,10)	17 (13,28)
II (p24Ag+)	5 (4,8)	22(18,34)
III (ELISA+)	3 (2,5)	25 (22,37)
IV (Western Blot +/-)	6 (4,8)	31 (27,43)
V (Western Blot +, p31-)	70 (40,122)	101 (71,154)
VI (Western Blot +, p31+)	open-ended	

**This table is reused with permission. All rights reserved.*

In the same paper we also show that the days since infection as estimated by our tool Poisson Fitter correlate quite well with the Fiebig stages and that the two are effectively equivalent, with the added advantage that the estimated days are a quantifiable measure that we can use for graphing purposes and in statistical analyses. Like with any theoretical method, exceptions can still occur: for example, CH1244, who was at Fiebig stage IV, was estimated to be only 12 days post infection by Poisson Fitter due to the highly homogenous viruses in each T/F population.

T/F. Transmission/Founder (T/F) viruses have been a convenient way to understand and explain the beginning of HIV infection; however, there remains controversy whether or not the methods used to determine whether a particular HIV sequence represents a true T/F variant is correct. The authors state “unambiguously” identify, but I do not think it is true. As noted in this paper’s Discussion, the rapid loss of T/F variants provide doubt to me that we can truly say which variants represent T/F variants anyway.

Answer: We agree and removed the word “unambiguously.”

In particular, it is not clear to me (and others) that the analysis methods used to characterize HIV viral populations that have already adapted to the new host, especially in the setting of burgeoning immune responses, can adequately characterize T/F strains. This is especially pertinent in this study because: 1) the whole premise of this study is that the T/F strains are true representations of the starting viral populations,

Answer: We thank the reviewer for the critique but would like to point out that the exact identification of T/Fs is really *not* the essential premise of this paper, and, as noted before, we could have in fact run the whole analysis without identifying any of the sequences as T/Fs. We made the best estimate we could of the T/Fs given the earliest time point data in order to

distinguish recombination that happened after transmission from possible recombination that happened in the donor prior to transmission, as is the case when recombinant variants are transmitted. The place this is most relevant is in the replacement time of the T/F lineages by recombinants, but we could in fact say that recombinant forms we observed at the first time point were fully replaced by new recombinants with a half time of 27 days and therefore, no matter whether the T/Fs strains we inferred can truly represent the starting viral populations our conclusions on viral recombination dynamics remain unchanged.

2) they only looked at people with heterogeneous viral populations (although I do not quite understand what the cut-offs were for deciding how diverse a viral population had to be to be eligible for the study), and 3) there was quite a few days between infection and sampling so adaptation should have at least taken place to some extent, even better engagement with CD4 and CCR5.

Answer: Again, these are valid points and they are discussed in detail in our two earlier papers (Keele et al. PNAS, 2008 and Lee et al. J Theor Biol 2009). We have added sentences in the main text to clarify that these methods are described in detail elsewhere. While we cannot prove that there was no selection prior to what we sampled (except for extreme purifying selection completely removing some deleterious mutations), the statistical evidence offered by our tool suggests that the mutations observed are consistent with a random accumulation of mutations at the first time point.

But just like for the identification of the T/Fs, this too is a really a side issue to the main objective of this paper, which is determining recombinants and how recombination affects viral evolution. In fact, we now show how the tool can be used in a setting where the infection was started by a single T/F, and there is indeed selection at play, using an example subject CH0505. To detect recombination in this setting, there has to be adequate time for enough diversity to accumulate to begin observing recombinants. By starting with subjects that already had already diversity at the earliest time point available, as it is the case in heterogeneous infections, we were able to track recombination in a longitudinal setting. Identifying the likely transmitted founders in the first time point provided a simple framework to enable this, but, as we have already observed, is not essential to running the recombination detection tool.

To this last point, I think the authors should use their tool to look at viral dynamics in monkeys who underwent dual infection with two known T/F to see if they get the same answers as they see in humans.

Answer: We have checked the literature and GenBank but could not find longitudinal sequences from monkeys deliberately infected with known multiple SIV or SHIV T/F viruses to do similar analysis (monkeys exposed to a quasispecies that have multiple infections from related strains do not provide any advantage over the human scenarios we are studying). Deliberate dual exposure can just result in a single virus emerging to establish the infection (Julg et al, Sci Transl Med. 9(408): eaao4235, 2017). Based on our experience with HIV and SIV genetic analysis, we anticipate that similar recombination dynamics are expected if the monkeys are infected with two or more different viruses. Nevertheless, we are beginning to apply RAPR to George Shaw's

many new SHIVs, and will look at the data over the next year or two. This work is currently in progress.

I also do not understand why lineages with <3 sequences were excluded from the analysis.

Answer: We have now clarified the reason in the text, namely that our statistical analyses are based on assumptions that break down when the clusters are small. In particular, we use a chi-square goodness of fit test, and such tests are unreliable for a sampling size of 3 or less.

Further, since this paper only evaluates people with heterogeneous T/F populations, then there may be a selection bias for these people. What is the reason for such diversity that persisted?

Answer: Regarding the selection bias, we have now clarified this point by adding the following sentence in the main text: “In order to have enough diversity to detect recombination from early infection, as matter of necessity, we focused on heterogeneously infected subjects. As a result, this study population may have biases relative to single T/F infections in terms of persistence of viral diversity. In addition to these 9 heterogeneous subjects, we also included 341 env sequences from the homogeneously infected subject CH0505, spanning day 19 through day 692 since infection.”

As for diversity, it is a known fact that the 2-arm race between the virus and the host immune pressure causes the viral population to continually diversify throughout the infection.

Was diversifying or purifying selection present; if so how was it measured? What if they applied RAP to the sequences generated from people with homogeneous viral populations? The hypothesis would be that no recombination would be detected, and I think that would be true, but maybe the investigators could check.

Answer: We have added an example of applying RAPR to homogeneous infections, and we now show how the tool can be informative in such scenarios. As we discuss in our simulated examples, a limit inherent to any recombination detection tool is that when multiple breakpoints fall in a region of homology between parents, the breakpoints will go undetected (see Supplementary Figs. 10 and 13). That’s why, when applied to the longitudinal samples from subject CH0505, homogeneously infected by a single T/F, recombination becomes apparent only after the viral population has sufficiently diversified, namely after 500+ days since the infection. However, we compared RAPR to other existing tools, and saw that our tool has considerably better sensitivity in terms of detecting recombination in a low diversity setting. Furthermore, RAPR detects many more recombinants than other existing tools even in this low-diversity setting of a subject in which only a single viral lineage is evident at the time of infection.

Immune selection and Viral load. The biggest weakness of this paper is the lack of measurement of immune responses (CTL, NAb, NK) to explain the observations presented. There is a very likely a selection pressure to drive recombination, or at least that hypothesis should be tested. The authors state this is likely but there is no follow up. Further, the authors discuss that immune pressures change (which is true) and are transient (which is usually not true). There should also be some evaluation for linkage of mutations with recombination. For example, if an escape

mutation in one variant recombines with an escape mutation in another variant. I would also expect some evaluation of at least in vitro replication capacity for the recombinants to better understand why the virus recombines so much. The authors discuss HLA haplotypes broadly in terms of viral loads and disease progression (protective and susceptible), but it is not clear how it relates to recombination, and it really cannot be assessed without actually looking for escape mutations. (For example, even neutral or susceptible HLA induce CTL that can cause escape mutations.)

Answer: We appreciate all those important points raised by the reviewer. We felt it was important to include an example illustrating recombination impacting selection for immune resistance. In response to this review we decided that our T cell example is difficult to follow, and so we have exchanged it for a neutralizing antibody escape example, where recombination is clearly shown to carry Ab resistance mutations forward in the complex quasispecies over time. This exchange also enabled us to illustrate how RAPR can be used to track recombination in an individual infected with a single virus, another point the reviewer addressed, and still remain within the space constraints of the journal.

There is some discussion about Nef escape mutations and maybe some escape mutations in Env from NAb, but how these escape mutations were determined is not clear from the paper. In particular, the determination of NAb breadth tells about escape of the whole virus but it does not tell us about which mutations were responsible for this escape and how these mutations influence the recombination observed.

Answer: We have now removed the discussion of the Nef escape mutations, and our new example specifically address the impact of the escape mutations.

The observations about VL are also interesting and how it may play in recombination. It is very clear from the literature that VL is directly related to CTL immune responses. It is also evident that VL and immune pressure are both related to recombination but the mechanisms and contribution of each are not assessed, since the effects of both (VL and immune response) are not independent of each other. There is some discussion of protective and susceptible HLA types in relation to VL and viral diversity, which has been documented before, but the study does not clearly outline how this is relevant to recombination.

Answer: We agree with the reviewer that this part is difficult to follow and have now removed it from the manuscript.

Reviewer #2 (Remarks to the Author):

The paper describes the development a new computer algorithm to genome recombination and its application to discern within patient (intra-quasispecies) recombination events during early phase HIV infection. Recombination is one of the important mechanisms whereby HIV (and

many other viruses) generate diversity that can impact on tropism and pathogenesis, immune escape and drug resistance. Studying recombination between highly diverse viruses (e.g. inter-subtype) is a relatively easy task. However, study of the virus population within an individual patient is complicated by the fact that the viruses are highly related, and breakpoint identification is difficult. The current method overcomes this by utilising runs testing to determine the randomness (or otherwise) of informative sequence states and from this the likelihood and location of recombination events. Applying this test to identify genetic recombination is not new, but its utilisation within a computational framework is, and it is this latter aspect that will render the analyses more applicable to large datasets and make it more accessible to the wider research community. Indeed its application for analysis of early stage HIV infection provides (subject to the caveats below) important insight into the role of recombination in early stage diversification of HIV and its possible impacts.

Points for clarification:

Line 280 – “RAP also assumes that parental ~~284~~ sequences are not sampled later than the recombinant”. Is this a fair assumption - surely, the whole point about viruses that from proviral reservoirs is that parental viruses can re-emerge? Given that it is impossible to sample the entire reservoir at any given time, it is feasible for viruses not sampled in the initial time points to ‘reappear’ later. This needs very careful consideration.

Answer: We agree with the point the reviewer makes. We have now added this caveat in the main text with the following: “However, one needs to be careful when interpreting results in that some earlier lineages may in fact be latent or escape sampling and then reappear at later time points. When in doubt, the user should consider multiple runs, with and without specifying sequence time points.”

Lines 370-388 and 390-415. These sections need toning down!! Whilst the authors show that sequence variants across key nef- CTL epitopes show plasticity through recombinant, they show no direct evidence that this has been driven by CTL escape or indeed allows this virus to escape those responses in this particular patient. Similarly the narrative around neutralisation also relies on surrogate markers of neutralisation and lacks appropriate necessary direct biological data. Without associated phenotyping data on neutralisation or CTL escape, these statements are all hypothesis and supposition. Intriguing perhaps, but they need validating.

Answer: We agree and thank the reviewer for the suggestion. We have now removed this part and only focused on our published results. We hope that the reviewer will find the new example with antibody escape from subjects Ch0010 and CH0505 much clearer.

Reviewer #3 (Remarks to the Author):

Summary

This paper has two goals, one is an empirical paper tracking HIV recombination and a second goal describing a new tool (software) to characterize recombination. Unfortunately, the two goals

get very confounded in this paper. On the second goal, the paper fails for a number of reasons. First, to introduce new software in an analysis setting like this, it is customary to evaluate the software, ideally with simulated data to show it is performing as expected. The authors do this to a certain extent, but they seem to do it as an after thought. In fact, the methods are actually described in the results section. They conclude from this effort that their approach underestimates recombination, but do not report on type I or type II error rates. Second, it is also appropriate to compare the new tool to existing tools to see if there is, indeed, better, worse, or similar performance. There are no data presented in the paper that give the reader any confidence that this new tool does anything useful. The authors do compare their approach to a few other methods, but using the empirical data. They all give different results. Because the data are empirical, you don't know the truth (the true frequency of recombination, the actual parents, or the breakpoints). So all you can conclude is that different methods give different results. Importantly (and somewhat disturbingly), there is already a very heavily used tool called Recombination Detection Program (RDP) – compared to the authors' 'Recombination Analysis Program (RAP). The RDP program is now in its fourth version and across all four versions the software has over 3500 citations. It is very surprising that the RDP package is not even mentioned nor cited in this paper.

Answer: We had indeed run multiple comparisons to RDP3, and because it did not perform favorably when compared to RAPR, we had left it out. We have now added detailed comparisons with the latest version, RDP4, and with the methods included the suite.

The fact that the authors have such a very similar name is concerning.

Answer: We have now changed the name to RAPR (pronounced “rapper”).

Either way, this paper cannot move forward without a direct comparison to RDP.

Answer: We thank the reviewer for the suggestion and as stated above, we have now included extensive comparisons and discussed the results in details and showed the comparison results in Supplementary Table 2.

While the authors set up an interesting experiment with some wonderful data, the results are still ambiguous because they do not actually know the true number of recombination events nor the breakpoints. So they cannot accurately calculate type I and type II error rates in RAP. The reader is then left with yet another method that claims to detect recombination. While the authors go through significant effort to generate simulated data, it is underutilized in terms of testing the method. They should report false positive rates and false negative rates and compare to other approaches (especially, RDP).

Answer: In the setting of homogeneous infections that we are considering, none of the methods we tried had a measurable Type I error rate when running on simulated datasets. We explained this by adding the sentence “Similar results were found when running all tools on simulated datasets that contained both mutated descendants and recombinants; in this scenario there were no false positives (Type 1 error) detected by any of the above tools, including RAPR”, but we now discuss how the Type II error rates distinguish them (Supplementary Table 2).

In the end, the manuscript is exceptionally long and cumbersome. It has methods in the results and goes back and forth between the biology and the recombination detection method. The work would benefit significantly from a partitioning of the software as its own paper with appropriate validation and comparison. Then an empirical paper on the application to this interesting HIV data set.

Answer: We thank the reviewer for the insightful comments. We have addressed the parts that are cumbersome and shortened them to allow for a smoother read. In addition, we are now including detailed comparisons of our tool with RDP4 both on the empirical data as well as simulated datasets where we know exactly how many and which recombinants have been introduced. We show that RAPR performs better than all other tools in RDP4 as well as other tools not included in RDP4 by showing that it detects considerably more recombinants and breakpoints, that are known to be valid in the simulation setting, and that these differences are particularly striking in low diversity settings, which is the setting that we want to highlight in this study. We explored both false positive (discussed in the text and above) and false negative rates (Supplementary Table 2). False negatives are complicated by regions of high homology limiting detection, an issue we discuss extensively in the text. We further motivate the importance of our tool by showing that being able to detect recombination in homogeneous infections (i.e. low diversity settings) has important repercussions on viral evolution, and that these kind of analyses bring to light the role of recombination in the diversification and immunogenesis of the viral population. Finally, our tool has more features that are useful in longitudinal studies, particularly regarding accounting for time of sample and tracking descendants of recombinants, which are also discussed in the text.

Other Issues

Pg 4, line 83 ‘between different studies’ should be ‘among’

Answer: Thank you, this has now been corrected.

Pg. 5, line 111 ‘SplitsTree provides a network’ is one of many software packages that estimate and/or visualize gene genealogies as networks.

Answer: We agree, it was just meant as an example, the sentence has now been removed.

Pg. 5, line 116 ‘Defining the frequency of recombinants is particularly important in phylogenetic ...’ Identifying the endpoints is also important.

Answer: We have now added “and identifying the breakpoints”

Pg 6, lines 118-122 – GARD does not provide a list of recombinants, but see RDP! This software implements a number of different approaches to detecting recombination, including a runs test (GENECONV). <http://web.cbio.uct.ac.za/~darren/rdp.html>

Answer: We are now including a direct comparison to RDP4 in the revised version, as well as all its companion tools. Please note that the test implemented in GENECONV is **not** the Wald-Wolfowitz Runs Test (see discussion below).

So contrary to the statement on page 6 that the runs test has not been implemented, it in fact has in RDP. This is the reference for the runs test applied to recombination: Padidam, M., Sawyer, S. & Fauquet, C. M. (1999). Possible emergence of new geminiviruses by frequent recombination. *Virology* 265, 218-225.

Answer: The above paper describes the recombination detection tool GENECONV, which we are now comparing to RAPR in Supplementary Table 2. According to the above paper, the statistical tests implemented by GENECONV are described in detail in Sawyer, 1989, which we read and concluded that it is **not** the same runs test implemented by RAPR. Our runs test is the one described in Bradley, J. *Distribution-free Statistical Tests*, Chapter 12, 1968, and in Takahata (1994), and is an exact test of the number of runs expected for each proposed choice of parents for a given sequence. We then correct for multiple testing using a permutation test.

Sawyer 1989 instead uses different statistics based on comparing pairs of sequences at a time: either the sum of squares of the lengths of the runs, or the maximum value of the length of the run. Both ignore mutations other than those displaying silent polymorphisms in the data set. The sampling distributions of these are determined directly through permutation tests. It is noteworthy that this method is useful for detecting recombination, but not for attributing parentage as our method attempts to do. To our knowledge, the runs test we implement here is not available elsewhere.

Page 10-11 lines 231-238, all this looks like methods to me, not results.

Answer: Thank you, we have moved those paragraphs to the methods.

Page 16, line 356 sliding window size of 20. How is this justified?

Answer: We thank the reviewer for bringing this up and we have added the following sentence into the Methods: “We initially explored various window sizes, between 10 and 50 nucleotides, and we saw that decreasing to smaller sizes from 20 nucleotides did not add any additional information, while increasing the size caused several potentially interesting regions to go undetected.”

Pg. 23, lines 520-521, how is homogenous versus heterogenous infection determined? Are there viral load, CD4+, etc. information on these individuals?

Answer: The method was first described in our paper Keele et al., 2008, where we determined the distinction between heterogeneous and homogeneous infections by looking at both phylogenetic trees and Highlighter plots from the first time point sequences. These are shown in our study for all our subjects (Fig. 1 and Supplementary Figs. 19-20), and one can see that both in the trees and in the highlighter plots, sequences tend to cluster into “groups” such that there is great within-group similarity and great across-group diversity. In addition, we calculate pairwise

Hamming distances (number of mutations) across all sequences and see that when there is only one lineage (homogeneous infection), the distribution is unimodal, whereas multiple lineages give rise to multimodal distributions. Such differences are clearly illustrated in Figure 2 from Keele et al., where panel A shows tree, highlighter plot, and Hamming distance frequency counts of a homogeneous infection, while the bottom panels show two heterogeneous infections:

*Copyright (2008) National Academy of Sciences, U.S.A.
 Reused with permission. All rights reserved.

Longitudinal viral load and CD4 T cell count data are available for all individuals described in the study are shown below.

Viral loads (viral copy numbers per millimeter) at different time points in all subjects during two years of follow-up

Subject	Screening	Enrollment	Week 1	Week 2	Week 3	Week 4	Week 8	Week 12	Week 16	Week 24	Week 36	Week 48	Week 60	Week 72	Week 84	Week 96
703010010	12,995	408,727	287,756	489,476	225,191	214,173	128,283	155,723	84,199	61,105	86,606	119,726	164,735	285,478	130,949	181,644
703010200	165,501	128,677	125,291	109,551	245,514	514,236	333,834	62,167	188,613	216,057	98,131	27,882	204,148	78,060	18,367	18,300
703010228	47,459	335,000	NA	162,481	NA	105,825	99,944	137,159	59,225	504,808	203,389	400,325	157,898	811,359	260,243	356,431
703010275	410,499	415,450	1,420,575	477,910	131,585	160,678	342,836	249,644	56,358	104,107	60,893	54,929	32,908	177,946	78,329	41,016
703011754	>750,000	642,000	696,822	167,665	738,636	1,010,000	673,400	487,805	265,000	229,061	443,000	430,000	280,000	799,000	409,000	226,000
703011244	>750,000	502,273	NA	NA	NA	236,260	>750,000	285,892	180,675	298,597	399,125	158,942	40,272	215,902	166,053	401,604
700010654	6,899,055	2,346,147	189,930	95,419	104,405	57,689	143,802	81,565	2,001,496	73,073	102,441	650	230,619	35,647	119,088	158,847
705010078	3,748,087	255,907	37,370	64,574	103,096	45,588	42,614	7,072	18,134	3,109	2,428	NA	2,877	NA	NA	NA
702010047	>750,000	331,625	>750,000	578,440	56,412	225,327	226,878	152,130	104,837	145,905	96,622	33,854	26,233	62,812	2,063	14,882
705010569	1,000,000	27,170	NA	5,601	3,278	3,571	552	3,250	408	324	3,044	NA	NA	NA	NA	NA
706010164	559	23,600	98,200	492,000	399,000	537,000	7,720	10,200	15,000	127,000	473,000	<400	819,000	653,000	258,000	NA
705010162	10,000,000	18,260	7,091	5,710	5,364	6,147	3,625	3,887	11,974	102,793	95,357	15,159	295,950	85,040	78,182	NA
705010107	10,000,000	44,516	22,377	8,920	2,505	846	419	336	113	232	314	217	713	440	972	949
703010256	62,060	254,060	17,990	46,342	29,682	35,328	31,400	23,626	13,484	NA	94,981	7,487	6,183	7,852	17,375	28,066
703010752	1,585,268	550,261	67,016	91,278	29,240	179,021	13,400	127,161	140,872	11,801	41,010	60,384	46,781	62,210	38,392	473,070
700010470	840,933	264,882	NA	NA	NA	27,991	27,869	39,257	10,613	28,456	18,160	58,848	10,104	NA	NA	NA
700010649	129,615	8,340	6,570	NA	NA	3,003	1,925	7,632	1,855	4,190	5,557	5,380	6,669	NA	NA	NA
703010131	411,873	437,369	9,366	7,764	10,655	53,169	45,697	15,424	20,806	50,709	27,446	44,488	95,637	41,231	19,168	17,807
700010040	2,197,248	298,026	NA	NA	NA	89,156	NA	17,587	12,836	29,453	7,580	NA	33,918	8,865	7,594	2,788
703011691	>750,000	252,474	59,014	407,416	151,611	99,229	378,069	217,670	129,243	111,839	138,577	85,121	110,039	32,645	220,253	86,314
703010694	2,809,692	243,636	36,531	20,192	25,055	17,834	35,551	4,763	184,388	141,113	68,489	49,117	162,550	NA	NA	213,417
700010058	92,581	394,649	NA	181,169	56,739	1,908	1,421	2,168	188	205	255	143	130	509	560	404
705010185	20,449	5,039,587	103,028	85,443	28,156	25,437	79,706	20,035	13,969	46,772	58,301	66,401	234,010	168,670	52,088	NA
705010198	10,000,000	301,401	106,441	87,085	15,036	4,525	2,433	410	2,326	578	NA	NA	3,476	NA	NA	NA
705010067	639,000	609	NA	337	963	2,124	<40	NA	<40	NA	NA	NA	<40	NA	<40	<40
700010077	179,031	144,145	37,560	NA	17,907	NA	NA	3,073	NA	1,680	NA	NA	NA	NA	9,165	NA
703010054	14,225	13,936	9,241	13,158	11,364	38,141	6,724	1,090	11,971	20,282	4,594	13,388	29,875	15,804	16,207	9,359
703010505	847,279	608,800	454,174	431,936	459,119	290,845	392,372	123,914	29,763	81,968	109,661	45,100	206,940	31,770	245,069	75,738
703010848	442,749	361,005	86,568	66,740	23,908	10,442	379,079	132,032	240,014	132,481	100,564	72,533	112,911	62,531	21,082	28,808
703010850	83,378	1,200,417	900,179	510,384	150,080	180,019	147,691	7,094	23,055	253,675	96,310	10,638	NA	NA	NA	31,866
703011432	407,760	345,260	167,056	27,471	222,275	278,110	469,426	181,038	86,584	595,809	136,384	NA	NA	NA	NA	NA
704010042	181,000	133,000	84,800	201,000	314,000	340,000	128,000	29,600	103,000	102,000	113,000	59,900	86,400	79,400	62,600	267,000
704010083	>750,000	6,440,000	735,000	708,000	78,700	125,000	588,000	517,000	174,000	293,000	404,000	198,000	162,000	133,000	NA	NA
704010236	>750,000	9,890	9,830	52,100	191,000	NA	90,200	179,000	255,000	209,000	657,000	NA	663,000	172,000	NA	NA
705010110	219,492	38,313	246,694	81,122	219,000	99,215	142,430	72,805	30,803	30,572	45,987	51,834	19,458	53,367	33,924	45,762
705010264	1,487,159	814,639	259,762	206,685	315,798	283,819	210,940	139,062	64,853	29,501	163,187	113,446	26,291	117,995	103,395	NA
703010159	17,805	73,453	587,457	570,786	7,101	13,566	9,397	71,474	12,815	4,926	3,722	9,822	12,120	NA	3,544	<400

NA: not available

CD4 T cell counts per microliter at different time points in all subjects during two years of follow-up

Subject	Enrollment	Week 1	Week 2	Week 3	Week 4	Week 8	Week 12	Week 24	Week 36	Week 48	Week 60	Week 72	Week 84	Week 96
703010010	202	NA	224	NA	NA	NA	NA	420	367	301	352	502	448	435
703010200	111	NA	NA	256	NA	NA	192	247	247	202	182	238	242	296
703010228	339	NA	NA	NA	NA	NA	253	356	232	283	275	196	190	169
703010275	160	NA	NA	318	NA	NA	250	287	310	336	344	341	321	372
703011754	589	NA	NA	NA	NA	NA	833	682	415	479	592	318	464	336
703011244	256	NA	NA	NA	NA	NA	241	246	223	330	248	193	250	200
700010654	NA	NA	NA	NA	NA	NA	435	379	489	417	512	374	235	269
705010078	251	365	NA	NA	NA	NA	NA	332	373	NA	481	NA	NA	NA
702010047	547	NA	NA	NA	NA	NA	NA	NA	594	473	450	626	378	247
705010569	387	NA	NA	NA	NA	NA	NA	561	858	NA	NA	NA	NA	NA
706010164	700	NA	NA	NA	NA	NA	669	449	459	377	475	443	445	NA
705010162	928	NA	NA	NA	NA	NA	NA	533	384	368	204	393	284	NA
705010107	428	NA	NA	NA	NA	NA	503	706	589	358	327	444	427	515
703010256	352	NA	NA	NA	NA	NA	747	600	597	601	639	659	531	NA
703010752	370	NA	NA	NA	NA	NA	577	501	487	441	446	508	472	382
700010470	324	NA	NA	NA	522	NA	679	554	503	383	329	NA	NA	NA
700010649	701	NA	NA	NA	NA	NA	509	NA	NA	NA	NA	NA	NA	NA
703010131	247	NA	NA	NA	NA	NA	395	310	300	298	315	340	266	214
700010040	929	NA	NA	NA	NA	NA	1,017	986	994	NA	1,396	882	941	972
703011691	451	NA	NA	NA	NA	NA	335	413	299	243	NA	345	315	272
703010694	488	NA	NA	NA	NA	NA	595	399	301	327	266	NA	NA	329
700010058	377	NA	NA	NA	554	NA	1,021	830	927	861	639	858	866	889
705010185	242	NA	NA	NA	NA	NA	NA	382	352	292	311	301	307	NA
705010198	599	NA	NA	NA	NA	NA	NA	673	NA	NA	1,115	NA	NA	NA
705010067	1,580	1,425	NA	NA	NA	NA	NA	1,684	1,400	1,692	1,557	1,551	1,917	1,313
700010077	806	NA	NA	NA	NA	NA	914	664	NA	NA	NA	NA	847	NA
703010054	593	NA	NA	NA	NA	NA	1,086	936	823	1,290	1,106	1,026	748	1,184
703010505	299	NA	NA	NA	NA	NA	288	338	295	431	244	255	378	347
703010848	624	NA	NA	NA	NA	NA	502	616	360	316	380	289	355	291
703010850	263	NA	NA	NA	NA	NA	506	427	479	540	NA	NA	NA	593
703011432	285	NA	NA	NA	NA	NA	445	378	183	NA	NA	NA	NA	NA
704010042	263	NA	318	NA	NA	350	NA	357	329	264	295	358	381	349
704010083	496	571	NA	NA	438	NA	NA	471	462	367	328	320	NA	NA
704010236	452	NA	NA	NA	NA	NA	NA	307	350	NA	348	59	NA	NA
705010110	602	NA	NA	NA	NA	NA	NA	433	359	379	409	493	425	421
705010264	524	NA	NA	NA	NA	NA	NA	606	334	353	313	278	242	NA
703010159	606	NA	NA	NA	NA	NA	463	413	448	658	506	NA	411	432

NA: not available

Reviewers' comments:

Reviewer #1 (Remarks to the Author):

Overall, I think the authors present a revised manuscript that is easier to read and understand. I also appreciate the use of the neutralizing antibody example to highlight their tool, which is much better than their previous CTL example. (This does raise a question whether CTL mutations impact the utility of the proposed tool but I doubt it.) I also like the new Discussion, although the overall importance of the method to scientific inquiry remains unclear. I still have some concerns.

There is not a 'true' biological dataset to evaluate their tool to substantiate that their new method is better than existing methods. The use of simulated data is nice but perhaps not biologically relevant, which would be important for a paper in such a high profile journal. The authors state that they will eventually test their program on viral data obtained from monkey studies where the monkeys are infected with two divergent viruses. This seems like a great test to me, and the publication of the tool that declares supremacy to other methods without these results seems premature. The authors state supremacy of their tool compared to others in the first sentence of the Discussion. Without clear biological data to test this hypothesis, I cannot certify that this is true.

The use of existing data from the CHAVI cohort is also nice in that it has been deeply characterized; however, there remain many issues for defining T/F variants for these persons, which is still not adequately addressed in the manuscript. (For example, a variant that is archived early infection could arise later and be called recombinant by the tool even though it is not or vice-versa.) The authors rely heavily on previous publications of this dataset to justify key points of their analyses, like that the circulating populations are homogeneous. The author's response says that these are not the central premise of the paper and yet these are the data used to evaluate their central premise. The lack of new primary data also dampens my enthusiasm for this paper.

I am also perplexed by the focus on the 'low diversity' setting. I understand how this makes the test less noisy, but this is rather rare among persons with HIV, except during acute infection, so I do not see how this helps the overall utility of the program.

Minor: The use of the word 'subject' is no longer politically correct, and I would advise using the term 'participant' and 'host'.

Minor: The sentence at lines 246-249 is not clear.

Minor: It is true that in general the genetic diversity of viral populations increases over a person's HIV infection, but not always, especially in regions like env or pol, where selective sweeps occur secondary to immune and drug pressure. This is also seen during viral rebound when ART is stopped. In fact, you can see a selective sweep in the env data from CH0505. This is also part of the problem for continuing to include a person who used ART in the analyzed dataset.

Minor: I found the explanation for the methods used to estimate duration of infection unnecessarily confusing. In particular, the data from CH1244 should raise considerable

concern with the veracity of the methods used.

Reviewer #2 (Remarks to the Author):

No additional comments

Reviewer #3 (Remarks to the Author):

The authors have done an extensive re-write and have accommodated my general concerns of a lack of comparison to other methods and lack of evaluation of their new approach with simulated data. The paper is written for HIV analysis only, which limits the breadth of applicability and therefore appeal of the paper. However, there are a lot of people running HIV sequence analyses who will find the approach useful. In lines 119-121 the authors classify RDP4 as a phylogenetic test. This is incorrect. The original RDP method is a phylogenetic test, but the RDP4 software implements a variety of tests, many of which are NOT phylogenetic, but rather runs tests similar to that proposed here. The overall change in focus to the recombination detection method and its performance with a biological example provides a much cleaner and more digestible manuscript. It flows much better and I appreciate the significant effort that went into changing things up to focus on the software and the presentation of performance results for the method.

The authors do not seem to make their code available, just a web version of the software. For academic software products, the standard convention is to make the code available via GitHub or some such mechanism. The end of line 860 is missing a period.

Overall, I found the revised draft a significant improvement and feel that the HIV community will take advantage of this new method to certain extent. However, the method seems to still be focused on data produced from Sanger sequencing and many HIV labs have moved to NGS data generation. It is not at all clear what the impact of many short sequences would be on this method's ability to detect recombination. I suspect it is not at all useful for NGS data.

Reviewers' comments:

Reviewer #1 (Remarks to the Author):

Overall, I think the authors present a revised manuscript that easier to read and understand. I also appreciate the use of the neutralizing antibody example to highlight their tool, which is much better than their previous CTL example. (This does raise a question whether CTL mutations impact the utility of the proposed tool but I doubt it.) I also like the new Discussion, although the overall importance of the method to scientific inquiry remains unclear. I still have some concerns.

Answer: We thank the reviewer for the positive feedback. We understand about the concerns, which hopefully we have fully addressed in the responses and revisions listed below.

There is not a 'true' biological dataset to evaluate their tool to substantiate that their new method is better than existing methods. The use of simulated data is nice but perhaps not biologically relevant, which would be important for a paper in such a high profile journal.

Answer: Simulated data is actually the best way to validate any detection tool because it allows full knowledge of the true recombinants and breakpoints that need to be detected. No biological dataset allows for that. Our simulated recombinant data was generated using real sequences as parentals, thus creating biologically realistic recombinants. Furthermore, by choosing parentals with different grades of homology, we were able to simulate different diversity settings and validate the tools in each scenario.

We have now added the following sentence to the manuscript to clarify the reason for simulated datasets:

“The best way to test this is using biologically sound simulated datasets where the exact recombinants and their breakpoints are known. To realize this, we randomly generated sets of 100 artificial recombinants with known crossover points, each from three different pairs of natural strains that carried a relative diversity of 0.6%, 1%, and 1.2% diversity respectively (see Methods).”

The authors state that they will eventually test their program on viral data obtained from monkey studies where the monkeys are infected with two divergent viruses.

We have been able to look into this, by examining data from a recently published real case scenario and found that the proposed experiment may not be feasible. The control group in Julg et al. (Julg et al., Sci Transl Med. 2017) had exactly the protocol the reviewer suggests: animals in the control group were infected by exposure to two divergent SHIV strains; in every case only one of the two established the infection. In the treatment groups, where an antibody was passively administered, the strain that was resistant to the antibody established the infection. In both the control (no antibody administered) and the antibody test groups, only

one strain or the other established the infection, not both. Using RAPR, we confirmed that there was no evidence of the second strain in the animals, and not even any trace of recombination between the two strains present at exposure or persisting in the infected monkeys.

This seems like a great test to me, and the publication of the tool that declares supremacy to other methods without these results seems premature. The authors state supremacy of their tool compared to others in the first sentence of the Discussion. Without clear biological data to test this hypothesis, I cannot certify that this is true.

Answer: As discussed above, with *any* biological data you would not know precise numbers or positions of breakpoints; even if two strains were successfully introduced into a monkey in future studies, and recombination occurred, the scientist studying the sequences would still be trying to estimate breakpoints, and there would be no known “ground truth” to compare computational tools that gave different results.

The point of a simulation is that we can create recombinants from real sequences, for which we know exactly where and how many breakpoints there are. All recombinants were generated from natural sequences to create a biologically realistic scenario. This is the only way to validate how accurate any recombination detection tool is at (1) correctly identifying recombinants, and (2) finding the breakpoints. We have tried to write this more explicitly in the text in order to make the strategy clearer to readers who are not used to thinking about simulations. We also now emphasize in the discussion that the improved sensitivity has been demonstrated in simulations with the following rephrasing:

"Compared to existing programs like RDP425 and GARD27, **simulations indicate that RAPR is more sensitive in detecting recombination events in low diversity settings ...**"

The use of existing data from the CHAVI cohort is also nice in that it has been deeply characterized; however, there remain many issues for defining T/F variants for these persons, which is still not adequately addressed in the manuscript. (For example, a variant that is archived early infection could arise later and be called recombinant by the tool even though it is not or vice-versa.)

Once again we would like to point out that RAPR can be used with or without T/F determination. For our paper we chose to run it with T/F determination because in our view it makes a better narrative and enables studying subject from early in infection. But recombinants can be found even without specifying T/F, one just has to sample far enough into the infection for adequate diversity to accumulate, to enable detection of recombination, as we did in CH505.

We agree that there can be ambiguity regarding whether the recombination event happened in the donor or in the recipient.

To further clarify this point, we have added the following text in the section “Study participants”:

“Previous studies have shown that 80% of sexually transmitted HIV-1 infections are initiated by a single T/F virus and only 20% are due to multiple, genetically distinct T/F viruses. In the latter case, due to the genetically distinct quasispecies coevolving in the host, it becomes easier to follow the history of recombination from the beginning of the infection. To this purpose, we distinguished participants productively infected with more than one virus (heterogeneous infection) from those infected with a single virus (homogeneous infection) by characterizing patterns of sequence diversity at the earliest time point (Fig. 1, Table 1, and Supplementary Figs. 1, 2) using statistical modeling, phylogenetic trees and highlighter plots, as previously described. However, it is important to note that the number of infecting strains, the incidence of superinfection, and the estimated number of days since infection play no role in the detection of recombination, except that we do not count the putative founder strains as recombinants since we are focusing on recombination in the recipient, rather than in the donor.”

The authors rely heavily on previous publications of this dataset to justify key points of their analyses, like that the circulating populations are homogeneous. The author’s response says that these are not the central premise of the paper and yet these are the data used to evaluate their central premise. The lack of new primary data also dampens my enthusiasm for this paper.

Answer: This is a misunderstanding, possibly because we have failed to highlight strongly enough the fact that while these participants have been previously described in other publications, **most of the sequences presented in this study had not been previously published**. In fact, there are 3,260 new and previously unpublished sequences in our study, which, upon publication, will be made publicly available on our LANL database.

To clarify, we have now added the following to the text:

"GenBank Access numbers for the newly obtained sequences in this study are: MF499156-MF502416. These 3,260 new sequences augment preexisting data to provide a unique data set, including extensive sets of longitudinally sampled sequences from multiple HIV infected individuals with complex multiple transmission events. All sequences from this paper, alignments and auxiliary data are also available in the HIV special interest alignments (https://www.hiv.lanl.gov/content/sequence/HIV/SI_alignments/datasets.html)."

I am also perplexed by the focus on the ‘low diversity’ setting. I understand how this makes the test less noisy, but this is rather rare among persons with HIV, except during acute infection, so I do not see how this helps the overall utility of the program.

Answer: We thank the reviewer for pointing this out. RAPR is uniquely sensitive in both low diversity and high diversity, while all other tools we have tested are only sensitive in high diversity scenarios.

When the parental strains are highly diverse, there will be many distinctive bases to use to infer recombination breakpoints (as in the case of inter-subtype recombination), and many tools are quite adept at picking up the recombination signal. However, in a low diversity setting, only a few bases may be indicative of a recombinant, making it harder to resolve parent and child differences and serial recombination events. RAPR is the first tool to address recombination in this scenario, and we have validated this through low-diversity simulated recombinants where RAPR picked up far more recombinants than pre-existing tools.

Minor: The use of the word 'subject' is no longer politically correct, and I would advise using the term 'participant' and 'host'.

Answer: We have replaced it with "participant".

Minor: The sentence at lines 246-249 is not clear.

Answer: We have edited the sentence to: "Despite the bias described above, which no recombination detection strategy can avoid, ..."

Minor: It is true that in general the genetic diversity of viral populations increases over a person's HIV infection, but not always, especially in regions like env or pol, where selective sweeps occur secondary to immune and drug pressure. This is also seen during viral rebound when ART is stopped. In fact, you can see a selective sweep in the env data from CH0505. This is also part of the problem for continuing to include a person who used ART in the analyzed dataset.

Answer: We agree and have edited the text to state that it **tends** to increase over time. Also, subject CH0505 was not on ART, and sequences collected from individuals after treatment were excluded from our analysis.

Minor: I found the explanation for the methods used to estimate duration of infection unnecessarily confusing. In particular, the data from CH1244 should raise considerable concern with the veracity of the methods used.

Answer: Our methods for the timing since infection have been described in detail in previously published work, where we show that it highly correlates with Fiebig stage. Like with all statistical tools, occasional deviations do happen and are to be expected, for example under strong selection. We thank the reviewer for pointing out that CH1244 is one of such exceptions, and we have now noted this in a footnote to Table 1.

Reviewer #2 (Remarks to the Author):

No additional comments

Reviewer #3 (Remarks to the Author):

The authors have done an extensive re-write and have accommodated my general concerns of a lack of comparison to other methods and lack of evaluation of their new approach with simulated data. The paper is written for HIV analysis only, which limits the breadth of applicability and therefore appeal of the paper. However, there are a lot of people running HIV sequence analyses who will find the approach useful.

Answer: RAPR is generally applicable to any situation with recombination and is particularly useful in a low diversity setting where other tools fail. We are HIV biologists, and this was the data and the problem to hand for us, but RAPR can in fact be run on sequences from other viral species. We now note this in the discussion as shown below and, furthermore, we are currently working on a manuscript where we apply RAPR to other viruses (work in progress).

“Finally we would like to note that intra-subtype recombination in HIV was first detected due to the high diversity of the virus, but it has also been observed in less diverse viral species such as hepatitis B viruses, enterovirus, and norovirus⁵¹⁻⁵³. Because of its higher sensitivity in low diversity scenarios, RAPR may prove useful in detecting recombination in other viral species where it has been more challenging to detect recombination.”

In lines 119-121 the authors classify RDP4 as a phylogenetic test. This is incorrect. The original RDP method is a phylogenetic test, but the RDP4 software implements a variety of tests, many of which are NOT phylogenetic, but rather runs tests similar to that proposed here.

Answer: That’s what we meant and we have now edited the text to clarify:

“While these tools range in methods and strategies, the RDP test itself is based on phylogenetic methods, and ...”

The overall change in focus to the recombination detection method and its performance with a biological example provides a much cleaner and more digestible manuscript. It flows much better and I appreciate the significant effort that went into changing things up to focus on the software and the presentation of performance results for the method.

The authors do not seem to make their code available, just a web version of the software. For academic software products, the standard convention is to make the code available via GitHub or some such mechanism.

Answer: We thank the reviewer for the suggestion, and we will be working on making the tool available on GitHub.

The end of line 860 is missing a period.

Answer: This has been corrected.

Overall, I found the revised draft a significant improvement and feel that the HIV community will take advantage of this new method to certain extent. However, the method seems to still be focused on data produced from Sanger sequencing and many HIV labs have moved to NGS data generation. It is not at all clear what the impact of many short sequences would be on this method's ability to detect recombination. I suspect it is not at all useful for NGS data.

Answer: NGS is becoming increasingly important, but there are different purposes to the kinds of data one collects with this method. NGS data is very helpful for studying variation in specific sites and regions within a patient, but not helpful for reconstruction covariation patterns in a protein, full length genomic sequences, or evolution within a patient. Such studies are still of great interest, particularly in the context of antibody-virus co-evolution.

However, in a slowly evolving virus at the population level, NGS data *could* be used to reconstruct an approximation of a full genome of a virus from a single individual (see for example Peccoud et al., 2018, doi: 10.1534/g3.117.300468), and in this setting RAPR could be used to detect recombination across a sampled population.